# Functional unknomics: Systematic screening of conserved genes of unknown function

João J. Rocha[1]☉, Satish Arcot Jayaram[1]☉, Tim J. Stevens[1]☉, Nadine Muschalik[1]☉, Rajen D. Shah[2], Sahar Emran[1], Cristina Robles[1], Matthew Freeman [1,3]*, Sean Munro [1]*

**1** MRC Laboratory of Molecular Biology, Cambridge, United Kingdom, **2** Centre for Mathematical Sciences, University of Cambridge, Cambridge, United Kingdom, **3** Sir William Dunn School of Pathology, University of Oxford, Oxford, United Kingdom

☉ These authors contributed equally to this work.
* matthew.freeman@path.ox.ac.uk (MF); sean@mrc-lmb.cam.ac.uk (SM)

**Data Availability Statement:** The Unknome can be viewed at http://unknome.org, with the entire database available to download as SQLite Version 3 files. Data from the functional screens are available

## Abstract

The human genome encodes approximately 20,000 proteins, many still uncharacterised. It has become clear that scientific research tends to focus on well-studied proteins, leading to a concern that poorly understood genes are unjustifiably neglected. To address this, we have developed a publicly available and customisable "Unknome database" that ranks proteins based on how little is known about them. We applied RNA interference (RNAi) in *Drosophila* to 260 unknown genes that are conserved between flies and humans. Knockdown of some genes resulted in loss of viability, and functional screening of the rest revealed hits for fertility, development, locomotion, protein quality control, and resilience to stress. CRISPR/ Cas9 gene disruption validated a component of Notch signalling and 2 genes contributing to male fertility. Our work illustrates the importance of poorly understood genes, provides a resource to accelerate future research, and highlights a need to support database curation to ensure that misannotation does not erode our awareness of our own ignorance.

## Introduction

The advent of genome sequencing revealed in humans and other species thousands of open reading frames that encode proteins that had not been identified by earlier biochemical or genetic studies. Since the release of the first draft of the human genome sequence in 2000, the application of transcriptomics and proteomics has confirmed that most of these new proteins are expressed, and the function of many of them has been identified [1]. However, despite over 20 years of extensive effort, there are also many others that still have no known function [2,3]. The mystery and the potential biological significance of these unknown genes is enhanced by many of them being well conserved and often being unrelated to known proteins and thus lacking clues to their function. Analysis of publication trends has revealed that research efforts continue to focus on genes and proteins of known function, with similar trends seen in gene and protein annotation databases [2,4,5]. This is despite clear evidence from studies of gene expression and genetic variation that many of the poorly characterised proteins are linked to disease, including those that are eminently druggable [6,7]. Indeed, it has long been argued that ignorance can drive scientific advance [8].

in the main text or the supplementary data sets S2 and S3 Data. Code for the functional assays is available at https://github.com/tjs23/unknome.

**Funding:** This work was supported by the Medical Research Council, as part of United Kingdom Research and Innovation (MC_U105178783 to SM and MC_U105178780 to MF). Work in MF's lab was supported by Wellcome Investigator Awards 101035/Z/13/Z and 220887/Z/20/Z. RDS was funded by the Engineering and Physical Sciences Research Council (EP/R013381/1) and by the Alan Turing Institute through a Turing Fellowship (TU/B/00006). The funders had no role in study design, data collection and analysis, decision to publish, or preparation of the manuscript.

**Competing interests:** The authors have declared that no competing interests exist.

**Abbreviations:** DUF, domain of unknown function; GO, Genome Ontology; GOA, Gene Ontology Annotation; MMAF, multiple morphological abnormalities of the sperm flagella; PCD, primary ciliary dyskinesia; RNAi, RNA interference; TRAP, translocon-associated protein; UPF, uncharacterised protein family.

This apparent bias in biological research toward the previously studied reflects several linked factors. Clearly, funding and peer-review systems are more likely to support research on proteins with prior evidence for functional or clinical importance, and individual perception of project risk seems likely to also contribute. In addition, scientific factors have been proposed, including a lack of specific reagents like antibodies or small molecule inhibitors, and a tendency to focus on proteins that are abundant and widely expressed and so likely to be present in cell lines and model organisms [4,7,9]. Finally, some genes may have roles that are not relevant to laboratory conditions [5].

Whatever the reasons, this inadvertent neglect of the unknown is clear and does not appear to be diminishing [9]. This has led to concern that important fundamental or clinical insight, as well as potential for therapeutic intervention, is being missed, and hence, the launch of several initiatives to address the problem. These include programmes to generate proteome-wide sets of reagents such as antibodies or mouse knock-out lines [10,11]. In addition, the NIH's Illuminating the Druggable Genome initiative supports work on understudied kinases, ion channels, and GPCRs [12]. There have been initiatives to develop new means to predict protein function or structure [13–17]. Finally, databases such as Pharos, Harmonizome, and neXt-Prot link human genes to expression and genetic association studies with the aim of highlighting understudied genes relevant to disease and drug discovery [18–20].

In this work, we have investigated directly the potential biological significance of conserved genes of unknown function by developing a systematic approach to their identification and characterisation. We have created an "Unknome database" that assigns to each protein from a particular organism a "knownness" score based on a user-controlled application of the widely-used Genome Ontology (GO) annotations [21,22]. The database allows selection of an "unknome" for humans, or a chosen model organism, that can be tuned to reflect the degree of conservation in other species, for example, allowing a focus on those proteins of unknown function that have orthologs in humans or are widely conserved in evolution. We use this database to evaluate the human unknome and find that it is shrinking only slowly. To assess the value of the unknome as a foundation for experimental work, we selected a set of 260 *Drosophila* proteins of unknown function that are conserved in humans and used RNA interference (RNAi) to test their contribution to a wide range of biological processes. This revealed proteins important for diverse biological roles, including cilia function and Notch pathway signalling. Overall, our approach demonstrates that significant and unexplored biology is encoded in the neglected parts of proteomes.

## Results

### Construction of an Unknome database

Much of the progress in understanding protein function has come from research in model organisms selected for their experimental tractability. Application of this research to the proteins of humans requires being able to identify the orthologs of these proteins in model organisms. Although it is not certain that orthologs in different species have precisely the same function, they generally have similar or related functions, implying that work from model organisms at the very least provides plausible hypotheses to test. Thus, our Unknome database was designed to link a particular protein with what is known about its orthologs in humans and popular model organisms.

A range of methods for identifying orthologs have been developed based on sequence conservation and although none are perfect, several achieve an accuracy in excess of 70%. We initially used the OrthoMCL database as it covered a wide range of organisms [23]. However, OrthoMCL was not being updated, and so the current Unknome database is based on the

PANTHER database (version 17.0) which covers over 143 organisms, is currently in continuous development, and has a good level of sensitivity and accuracy [24–26].

The heart of the Unknome database has been the development of an approach to assigning a "knownness" score to proteins. This is not trivial and is inevitably a somewhat subjective measure. Definitions of "known" range from a simple statement of activity to an understanding of mechanism at atomic resolution, and even well-characterised proteins can reveal unexpected extra roles. Thus, we designed the database so that the criteria for knownness can be user-defined, as well as having a default set of criteria. The GO Consortium provides annotations of protein function that are well suited to this application. Firstly, GO annotation is based on a controlled vocabulary and so is consistent between different species, and secondly, it is well structured thus allowing a user to apply their own definition of knownness.

The Unknome database combines PANTHER protein family groups (which we term "clusters") with the GO annotations for each member of the cluster. This includes annotations from humans and the 11 model organisms selected by the GO Consortium for their Reference Genome Annotation Project. The sequence-similar protein clusters (primary PANTHER families) not only contain orthologs, but also recent paralogs: duplications within individual species or lineages. The knownness score for each protein is calculated from the number of GO annotations it possesses.

It is important, however, to recognise that GO annotations do not all have equal evidential value, but they helpfully include an evidence code that indicates the type of source it is derived from. The Unknome database allows users to make use of this in generating a knownness score with an option to apply greater weight to annotations that are more likely to be reliable, such as those from a "Traceable Author Statement" rather than those "Inferred from Electronic Annotation" (Fig 1A and S1A Fig). In addition, weighting allows the selection of annotations most relevant to function. For instance, a protein's subcellular location is often included in its GO annotation, but this may not helpfully restrict the range of possible functions, so the database provides the option of excluding it when calculating a knownness value. The final knownness score of a cluster of proteins is set as the highest score of a protein in the cluster (Fig 1B).

The Unknome database is available as a website (http://unknome.org) that provides all protein clusters that contain at least 1 protein from humans or any of 11 model organisms (Fig 1C). The clusters can be ranked by knownness, and the user can modify this list so as to include only those proteins that are present in a particular combination of species, such as human plus a preferred model organism (Fig 1D). For each protein family, the interface shows the orthologs in its cluster and how the knownness of the cluster has changed over time (Fig 1C). These design principles maximise the versatility and power of the Unknome database as a tool for researchers from different biomedical fields.

## Validation of the Unknome database

To confirm that the Unknome database was accurately capturing current understanding of protein function, we ranked the 7,515 clusters of orthologs and paralogs that contain at least 1 human protein. Reassuringly, the top 10 scoring proteins have well-known roles in development and cell function (Fig 1E). In contrast, proteins containing one of the "Domains of Unknown Function" defined by the Pfam database were concentrated at the bottom of the range (S1B Fig). Clusters with a score of 1.0 or less correspond to 18.3% of all clusters but to 36% of the domains of unknown function (DUFs) and 59% of the related uncharacterised protein families (UPFs). The exceptions were typically multidomain proteins of known function that contain 1 domain whose role is unclear. Finally, the total number of PubMed citations for

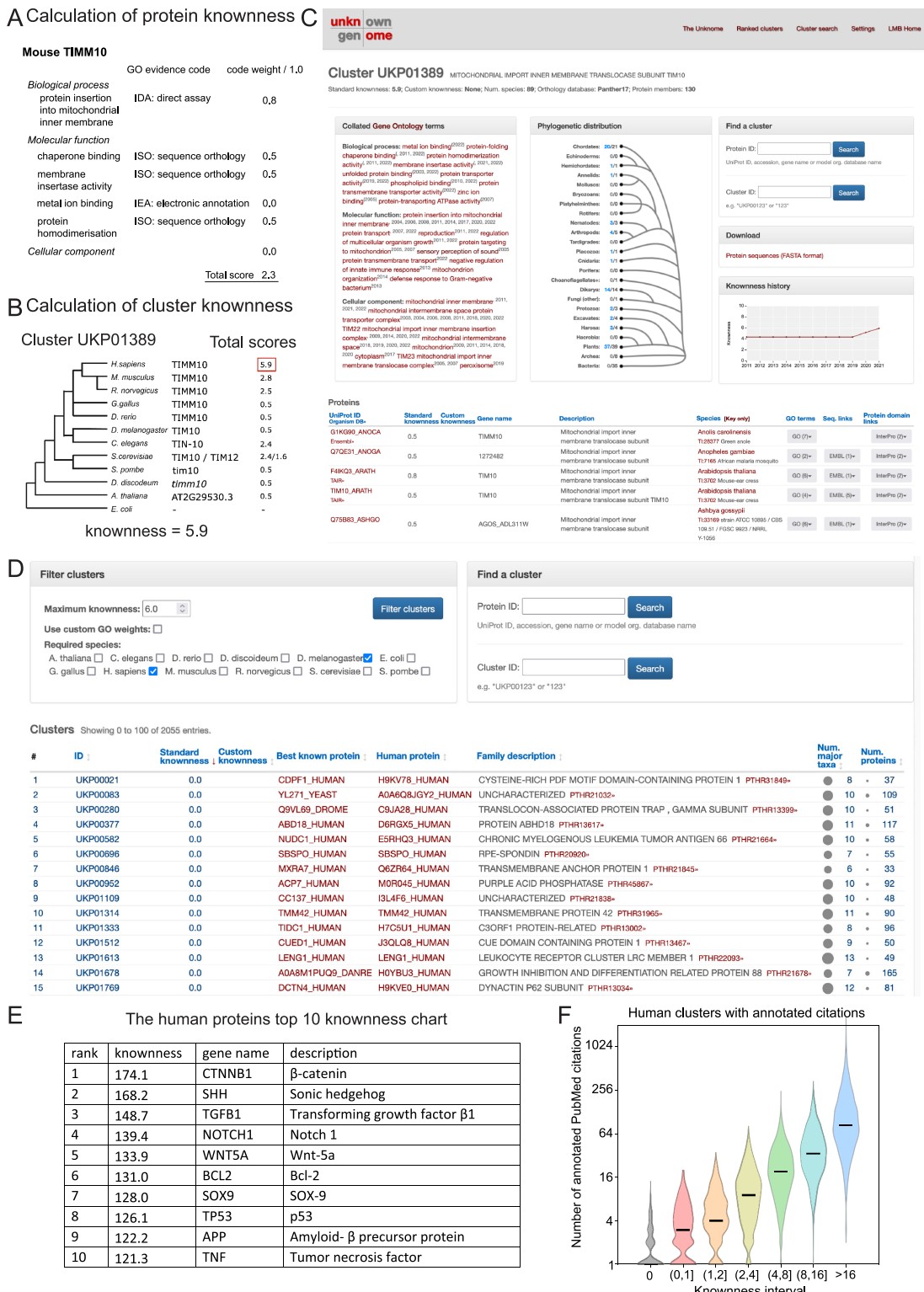

**Fig 1. The Unknome database.** (A, B) Calculation of a knownness score for a cluster of orthologs based on the highest score in the cluster. Illustrated with a cluster corresponding to a subunit of a mitochondrial inner membrane translocase; (A) shows the GO annotations for mouse TIMM10, and derivation of a score based on the number of annotations weighted for their confidence, while (B) shows the scores for all the members of the cluster containing TIMM10 (UKP01389), with the highest score of a member being the knownness of the cluster. (C) The Unknome database contains information for each cluster showing its

distribution across species, links to information for the protein from each species, and the change in knownness over time—as illustrated for cluster UKP01389. (D) User interface to list clusters from a user-selected set of model organisms by the knownness of the cluster. The list indicates the best-known member of the cluster and the human member(s) of the cluster. (E) The 10 best known protein clusters, showing the best-known human gene in each. (F) Plot of the number of PubMed citations in the Uniprot comments section for human-gene containing clusters in the indicated range of knownness. The data underlying the plot can be found in S1 Data. GO, Genome Ontology.

each protein shows a good correlation with the knownness scores from the database (Fig 1F). Overall, we conclude that the calculated knownness score provides a useful means to identify proteins of unknown function.

## The change of the Unknome over time

Unlike most databases, the Unknome will shrink over time. The knownness scores for clusters containing human proteins have increased across the whole range of proteins, but the proportion with a knownness score of 2 or less has declined from 43% to 23% over the last 10 years, with the decline being less in nonhuman model organisms (Fig 2A and S2A Fig). This slow progress is unlikely to represent a deficit in GO annotation which is kept up to date, but rather that human genes and proteins are much more likely to have been published on in the last 12 years if they are in clusters that were already well known at the start of this period (Fig 2B and S2B Fig). Consistent with this, knownness increases more rapidly over time for genes that were already well annotated (S2C Fig). These observations provide further support to the notion that research activity tends to focus on what has already been studied in depth [2,4,27]. There are 750 human clusters whose knownness was zero 12 years ago but has since increased to above 2. The GO terms most enriched in this set are mostly associated with cilia, reflecting recent acceleration of progress in studying this large and complex structure that is absent from some model organisms such as yeast (Fig 2C). Consistent with this, the less known human genes tend to be less likely to be conserved outside of vertebrates, and generally have fewer orthologs, suggesting that progress has been hampered by there being fewer orthologs that could be found by genetic screens in non-vertebrates (S2D and S2E Fig). Interestingly, the most highly known proteins are also less likely to be conserved outside of metazoans, reflecting the fact that many are involved in important developmental pathways or signalling events relevant to multicellularity (S2D Fig). However, of the 1,606 human-containing clusters with a current knownness score of less than 2.0, 68% are detectably conserved outside of vertebrates and 45% are conserved outside of metazoans (Fig 2D). Interestingly, no one model organism contains all of these, indicating that each has a role to play in illuminating the human unknome.

## Functional unknomics in *Drosophila*

To test the value of the Unknome database, and to pilot experimental approaches to studying neglected but well-conserved proteins, we selected a set of unknown human proteins that are conserved in *Drosophila* and hence amenable to genetic analysis. *Drosophila* also tends to lack partial redundancy between closely related paralogs, as in humans this arose in many gene families from the 2 whole-genome duplications that occurred early in vertebrate evolution [28]. A powerful approach to investigating gene function in *Drosophila* is to knockdown its expression with RNAi and assess the biological consequences [29,30]. We thus determined the effect of expressing hairpin RNAs to direct RNAi against a panel of genes of unknown function.

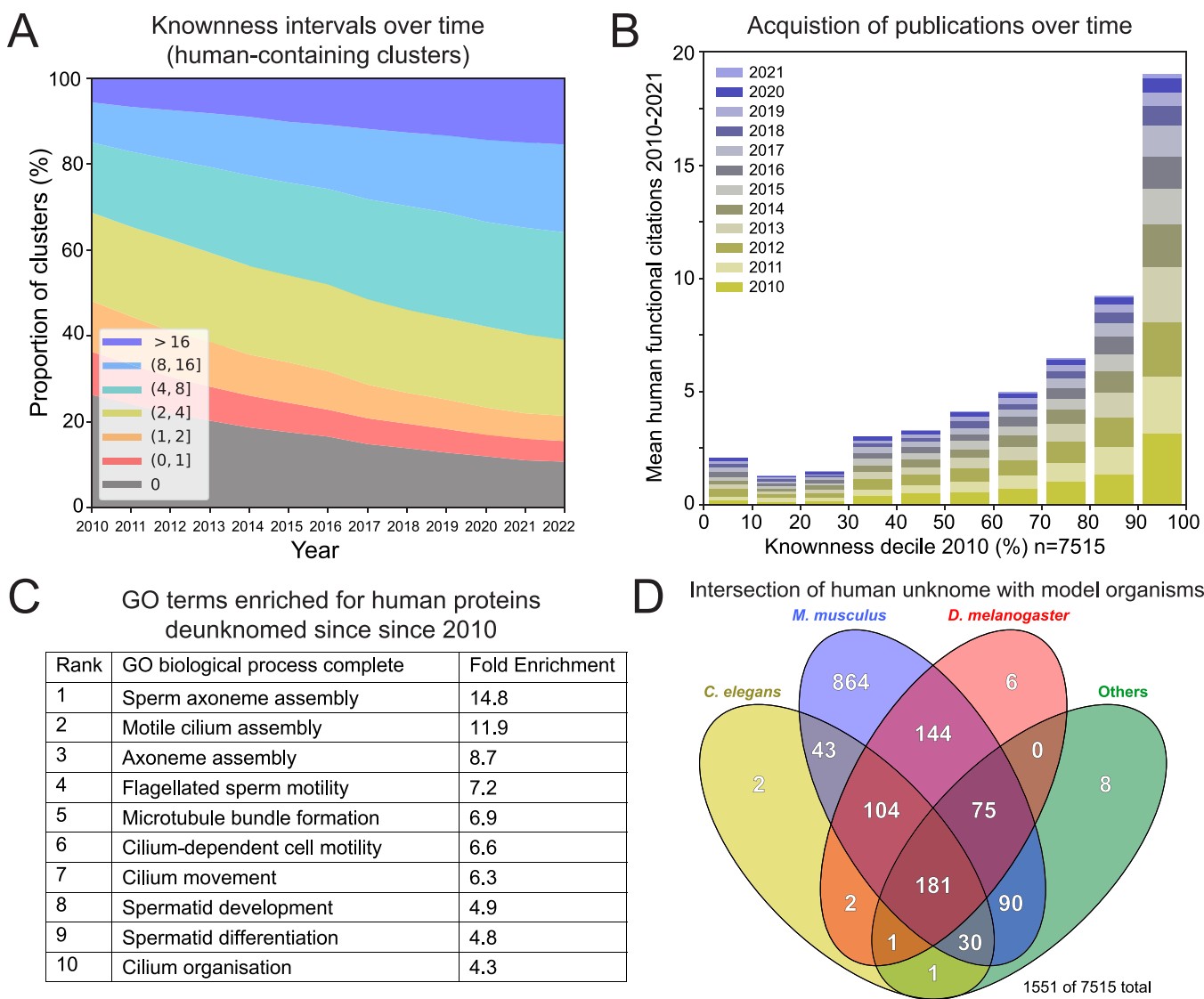

**Fig 2. Analysis of trends in knownness.** (A) Change in the distribution of knownness of the 7,515 clusters that contain at least 1 protein from humans. (B) Mean number of publications added each year since 2010 to the UniProt entry for the human protein in each of the 7,515 clusters that contain at least 1 human protein, ranked into deciles based on knownness at 2010. Where there was more than 1 human protein in the cluster, their publications were summed. The best-known clusters in 2010 received the most publications in subsequent years. (C) The 10 largest GO term enrichments for the 753 human proteins from clusters whose knownness has increased from 0 in 2010 to 2.0 or above by 2022. When there was more than 1 human protein in the cluster, a single one was used chosen by alphabetical order to avoid bias. GO enrichment analysis used ShinyGO [112]. (D) Venn diagram showing the distribution of genes from the indicated species in the 1,551 clusters of knownness <2.0 and which contain at least 1 human protein. Not shown are the 55 clusters that appear only in humans. The data underlying the graphs shown in the figure can be found in S1 Data. GO, Genome Ontology.

We initially selected all genes that had a knownness score of ≤1.0 and are conserved in both humans and flies, as well as being present in at least 80% of available metazoan genome sequences. Of the 629 corresponding *Drosophila* genes, 358 were available in the KK library that was the best available genome-wide RNAi library at the time (S1 Table) [31]. This, and other RNAi libraries, have been used for several genome-wide screens for phenotypes readily analysed at large scale, but had not been used for the screens that we applied [31]. These KK library stocks were crossed to lines containing Gal4 drivers to express the hairpin RNAs in

either the whole fly or in specific tissues. After testing for viability, the nonessential genes were then screened with a panel of quantitative assays designed to reveal potential roles in a wide range of biological functions. These include male and female fertility, tissue growth (in the wing), response to the stresses of starvation or reactive oxygen species, proteostasis, and locomotion. The results of these screens are discussed below.

## Unknown genes have essential functions

To determine if the genes were required for viability, a ubiquitous GAL4 driver was used to direct RNAi throughout development (daughterless-Gal4). For 162 of the 358 genes, the resulting progeny showed compromised viability with either all (lethal) or almost all (semi-lethal) failing to develop beyond pupal eclosion, suggesting that these genes are essential for development or cell function (S1 Table). However, it was subsequently reported that in a subset of the lines in the KK RNAi library, the transgene is integrated in a locus (40D) that itself results in serious developmental defects when the transgene is expressed with a GAL4 driver [32,33]. Following PCR screening, we removed all of the stocks that had this integration site, all but one of them having been lethal in the initial screen. For the remaining 260 genes, the stocks used the alternative integration site which is not problematic, with KK stocks having been used successfully in a range of different screens [29,34]. For these, the RNAi compromised viability in 62 cases (24%). In considering the results from RNAi screens, one must always be mindful of off-target effects, and in *Drosophila*, the possible effects of variability in genetic background and conditions of rearing and maintenance. Nonetheless, of these 62 genes, 12% were also identified in a recent genome-wide screen of genes required for viability of S2 cells; in contrast, only 4% of the 198 nonessential genes were hits in the S2 cell screen [35]. The S2 study estimated that 17% of genes known to be essential in flies are also essential in S2 cells, and it is likely that using RNAi to knockdown gene function underestimates lethality. Our screen in whole organisms reveals that, despite several decades of extensive genetic screens in *Drosophila*, there are many genes with essential roles that have eluded characterisation.

Of course, there is more to life than being alive. We therefore subjected the 198 apparently nonessential genes to a range of phenotypic tests to determine if they had detectable roles in a wide range of organismal functions. On the grounds that the long history of *Drosophila* genetic screens may have saturated the discovery of mutants with easily detectable phenotypes (mostly developmental defects), we targeted our search to nonstandard and quantitative phenotypes that are harder to assess. In practice, this meant designing phenotypic screens that were more complex than normal. Our hope was that this would identify a larger proportion of genes that had not been hit in more standard *Drosophila* screens. The results of these function screens are described below, followed by a validation of selected hits, with the screening data provided in S2 and S3 Data and the results summarised in S2 Table.

## Contribution of unknome genes to fertility

To test fertility, specific GAL4 drivers were used to knockdown the set of 198 unknown genes in either the male or female germline. Even with collecting data for multiple flies per gene, the resulting brood sizes showed some variability, as expected for a quantitative measure of a biological process. Thus, for all our assays, we needed to determine if outliers had a phenotype that exceeded to a statistically significant degree the variation intrinsic in the population. To do this, we used statistical tests based on 3 steps. First, we performed a regression on the replicate data for each gene to estimate its parameters and standard errors within the assay. Next, an outlier region was determined by fitting the parameter estimates for all analysed genes to a normal distribution, which was then used to define a boundary for outliers. Finally, for each

gene, we tested the hypothesis that it falls within the outlier boundary. This approach is summarised in the Methods and described in detail in the Supporting information (S1 Text). To display the data from the fertility tests, mean brood sizes obtained from RNAi-treated males was plotted against those obtained from RNAi-treated females for each gene (Fig 3A). Several of the RNAi lines gave a substantial reduction in brood size that was sex specific and highly statistically significant.

**Female fertility.** Two genes gave a partial, but significant, reduction in female brood size. During the course of our work, a mouse ortholog, MARF1, of one of these hits, CG17018, was identified in a genetic screen as being required for maintaining female fertility, apparently by controlling mRNA homeostasis in oocytes [36,37]. A recent study of CG17018 has confirmed that it is indeed required for female fertility in *Drosophila*, despite lacking some domains present in MARF1. Its appearance as a hit in our screen is therefore an encouraging validation of the approach [38]. The other gene, CG8237, has not previously been linked to fertility, but has a mammalian ortholog (FAM8A1) that has been recently proposed to help assemble the machinery for ER-associated degradation (ERAD) and so may have an indirect effect on oogenesis [39,40]. We selected CG8237 for validation by CRISPR/Cas9 gene disruption as described below.

**Male fertility.** Seven genes showed near complete male sterility, with 5 further genes giving a statistically significant reduction in brood size. In humans, male sterility is one of the symptoms associated with primary ciliary dyskinesia (PCD), a disorder affecting motile cilia and flagella. While our analysis was in progress, exome-sequencing allowed the identification of many new PCD genes [41,42]. Interestingly, 5 of the genes identified in our assay are homologs of human PCD genes (Fig 3B), of which CG5155 (ARMC4) and CG31320 (DNAAF5) have since been shown to be required in *Drosophila* for male fertility [43,44]. All of these genes comprise, or help assemble, the dynein-based system that drives the beating of cilia and flagella. In addition, human orthologs of 2 of the semi-sterile hits in the Unknome screen have been found to be mutated in related familial conditions. CFAP43 (orthologous to CG17687) is mutated in patients with multiple morphological abnormalities of the sperm flagella (MMAF), and CFAP52 (orthologous to CG10064) is mutated in laterality disorder, a condition caused by defects in ciliary beating during development [45,46]. A further semi-sterile hit, CG14183, is an ortholog of DRC11, a subunit of the nexin-dynein regulatory complex that regulates flagellar beating in *Chlamydomonas* [47]. These findings prove the value of the Unknome database approach to identifying new genes of biological significance and validate the RNAi-based screening approach.

Of the 4 remaining genes that showed male fertility defects, CG11025 is now only partially unknown as its human ortholog (UBAC1) is a non-catalytic subunit of the Kip1 ubiquitination-promoting complex, an E3 ubiquitin ligase [48]. CG11025 was recently identified in a genetic screen for defects in ciliary traffic and found to be required for fertility [49]. However, the other 3 genes, CG8135, CG6153, and CG16890 (orthologous to LMBRD2, PITHD1, and FRA10AC1), remain poorly understood in any species. They are less likely to be flagellar components as they are not predominantly expressed in testes and, as described below, 2 were selected for validation by CRISPR/Cas9 gene disruption, along with CG10064 whose ortholog CFAP52 is mutated in laterality disorder.

## Contribution of unknome genes to tissue growth

To test the unknome set of genes for roles in tissue formation and growth, we examined the effect of knocking them down in the posterior compartment of the wing imaginal disc and comparing the area of the posterior compartment of the adult wing to that of the control

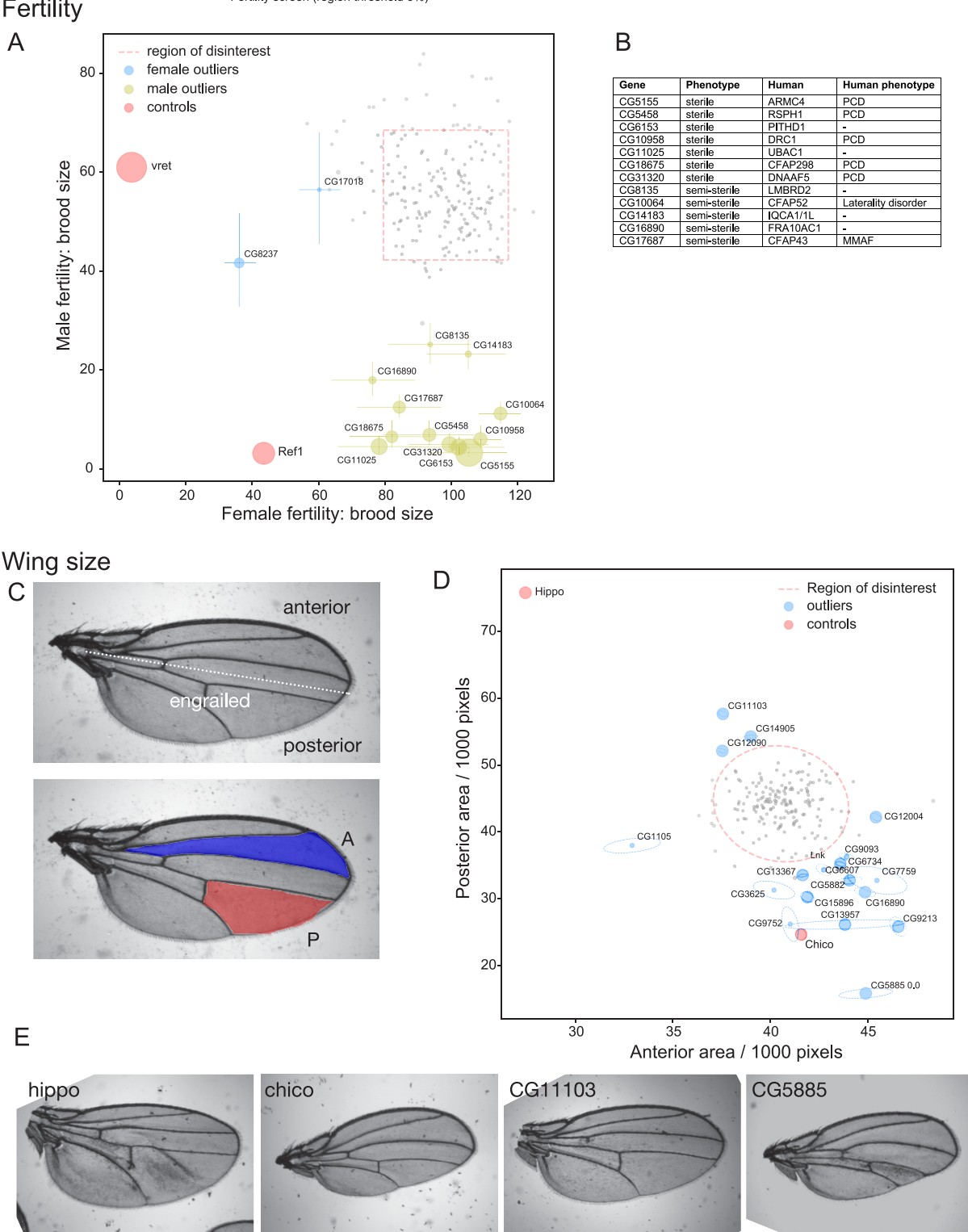

**Fig 3. Testing of the unknome set of genes for roles in fertility and wing growth.** (A) Plot of brood sizes obtained from matings in which each gene was knocked down in either the male or female germline. Dotted lines indicate outlier boundaries, with the genes named being those whose position outside of the boundary is statistically significant, error bars show standard deviation, and the size of the circles is inversely proportional to the *p*-value. Controls: Vret is involved in piRNA biogenesis and affects female fertility [113], and Ref1 is an essential protein predicted to be involved in RNA export [114], and affects both males and females. (B) Summary of the significant hits from the test of male

fertility, showing the human ortholog and the phenotype reported for patients with loss of function mutations (PCD, MMAF). (C) Adult wing illustrating the posterior domain that expresses engrailed during development and hence the engrailed-Gal4 driver used to express the hairpin RNAs. Also shown are the intervein areas measured to assess tissue growth in the anterior and posterior halves of the wing. (D) Plot of the mean area of the anterior and posterior intervein areas as in (C) for flies in which each gene was knocked down by RNAi in the posterior domain (pixel dimensions 2.5 μm × 2.5 μm). Errors are shown as tilted ellipses with the major/minor axes being the square roots of the eigenvectors of the covariance matrix. Dotted lines indicate the outlier boundary, with the genes named being those whose position outside of the boundary is statistically significant, with the size of the circles being inversely proportional to the *p*-value. The genes Hippo (growth repressor) and Chico (growth stimulator) were included as controls. (E) Representative wings from flies expressing hairpin RNA for the indicated genes in the posterior domain. Hippo and Chico are controls as in (D), with CG11103 and CG5885 showing an increase or decrease in the posterior domain, respectively. The means and variances used for the graphs shown in the figure can be found in S2 Data with the data points in S3 Data. MMAF, multiple morphological abnormalities of the sperm flagella; PCD, primary ciliary dyskinesia; RNAi, RNA interference.

anterior compartment (Fig 3C), a method previously used to detect effects of a range of different genes [50,51]. As controls, we used Hippo, a negative regulator of tissue size, and Chico, a component of the PI 3-kinase pathway that stimulates organ growth [52,53]. Knockdown of 3 of the unknome genes in the posterior compartment caused a statistically significant increase in its area (Fig 3D and 3E). These include CG12090, the *Drosophila* ortholog of mammalian DEPDC5, which was found to be part of the GATOR1 complex that inhibits the Tor pathway during the protracted course of our studies. Mutants in GATOR1 subunits promote cell growth by increasing Tor activity [54,55]. The other 2 are CG14905 and CG11103. CG14905 is a paralog of a testes-specific gene CG17083, and both are orthologs of mammalian CCDC63/CCDC114 that have a role in attaching dynein to motile cilia, although CG14905 seems likely to have additional roles as it is ubiquitously expressed [56]. CG11103 (TM2D2) encodes a small membrane protein that shares a TM2 domain with Almondex, a protein with an uncharacterised role in Notch signalling [57]. We therefore selected CG11103 for further validation by CRISPR/Cas9 as described below.

A larger number of genes caused a reduced compartment size when knocked down (Fig 3D). However, this could arise from a wide range of causes and so this is broad ranging assay for protein importance, and indeed mammalian orthologs of several of the stronger hits have been subsequently found to act in known cellular processes such membrane traffic (CG13957, the ortholog of human WASHC4), lipid degradation (CG3625/AIG1), or tRNA production (CG15896/PRORP). The strongest effect was seen with CG5885, an ortholog of a subunit of the translocon-associated protein (TRAP) complex that is associated with the Sec61 ER translocon [58]. TRAP's role is enigmatic and so it was also selected for CRISPR/Cas9 validation.

## Contribution of unknome genes to protein quality control

The removal of aberrant proteins is a fundamental aspect of cellular metabolism, and thereby organismal health, but it is a function that does not necessarily contribute substantially to well-screened developmental phenotypes. It also exemplifies our suspicion that a disproportionately high number of the unknome set of genes may be involved in quality control and stress response functions, which are likely to have been missed by many traditional experimental approaches. We therefore tested the unknome gene set for protein quality control phenotypes, using an assay based on aggregation of GFP-tagged polyglutamine, a structure found in mutants of huntingtin that cause Huntington's disease [59]. When this Httex1-Q46-eGFP reporter is expressed in the eye, the aggregates can be detected by fluorescence imaging (Fig 4A). The RNAi guides were co-expressed in the eye to knockdown unknome genes, and the number of polyQ aggregates quantified for 2 different size ranges. Although there was considerable variation in aggregate number, statistical analysis allowed the identification of clear outliers among the unknome RNAi set (Fig 4B). Most of the genes showing the largest increase

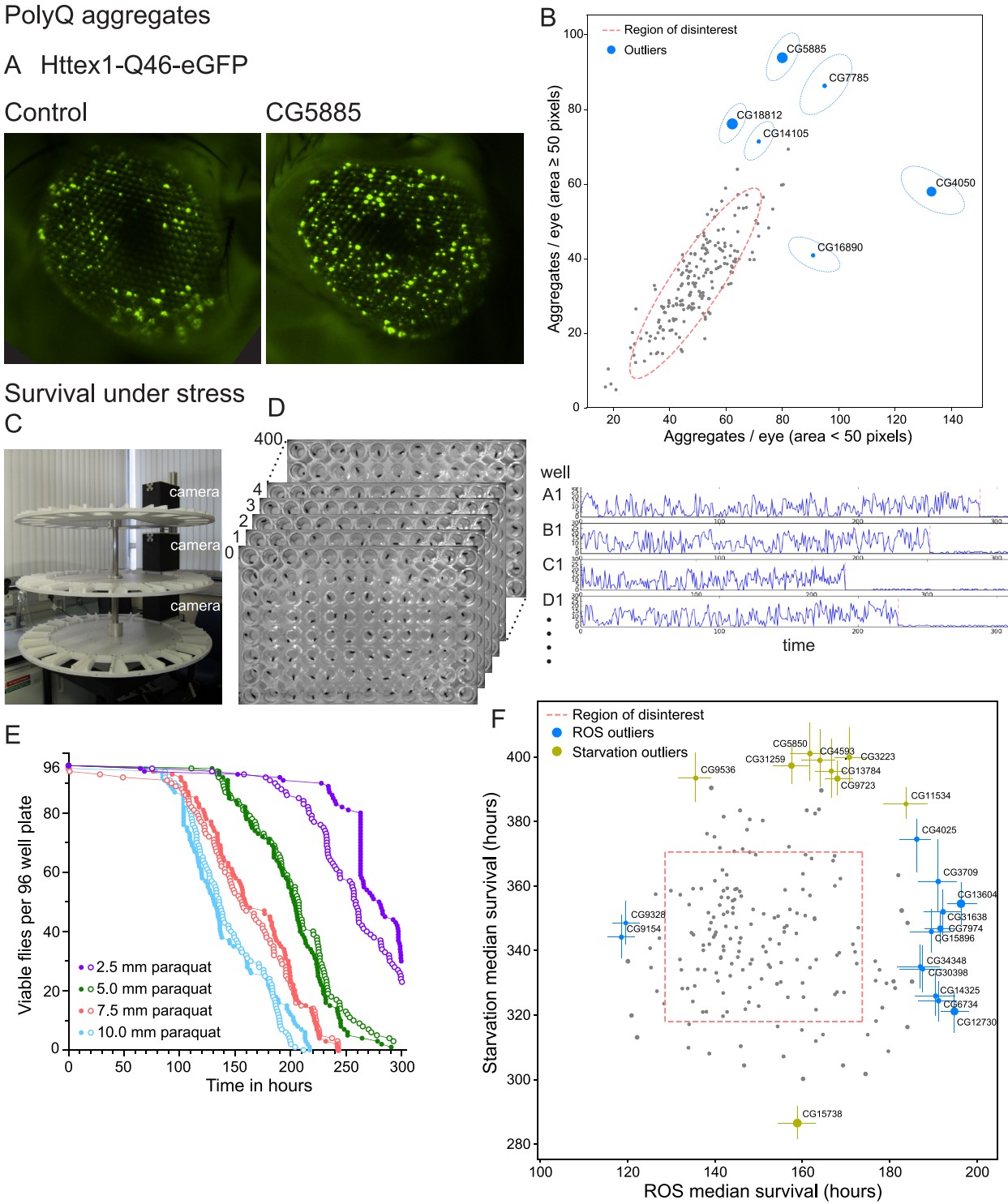

**Fig 4. Testing of the unknome set of genes for roles in quality control and responses to stress.** (A) Fluorescence micrographs of eyes from stocks expressing Httex1-Q46-eGFP along with either no RNAi, or one to the screen hit CG5885, both under the control of the GMR-GAL4 driver. The GFP fusion protein forms aggregates whose number and size increase over time. (B) Plot of the mean number of large ($\geq$50 pixels) or small (<50 pixels) aggregates of Httex1-Q46-eGFP formed after 18 days in flies in which the unknome set of genes has been knocked-down by RNAi (pixel dimensions 0.5 μm × 0.5 μm). Errors are shown as tilted ellipses with the major/minor axes being the square roots of the eigenvectors of the covariance matrix. Dotted lines indicate an outlier boundary set at 90% of the variation in the dataset, with the genes named being those whose position outside of the boundary is statistically significant with a $p$-value <0.05, with the size of the circles being inversely proportional to the $p$-

value. (C) Flywheel apparatus for time-lapse imaging of 96-well plates containing 1 fly per well. Each of 3 wheels holds 20 plates that rotate under a camera to be imaged once per hour. (D) Use of time-lapse imaging to assay viability: 96-well plates were imaged very hour and the movement between frames quantified for the fly in each well. Plots of movement size over time allow the time point for cessation of movement and hence loss of viability to be determined automatically. (E) Survival plots obtained from the flywheel for flies in 96-well plates with food containing the indicated concentration of oxidative stressor paraquat. Increased levels of the paraquat shorten survival times. Two independent 96-well plates are shown for each condition to illustrate the reproducibility of the assay. (F) Plot of the median survival time of fly lines in which the unknome set of genes has been knocked-down by RNAi and which were then exposed to paraquat to induce oxidative stress or were starved for amino acids. Dotted lines indicate an outlier boundary set at 80% of the variation in the dataset, with the genes named being those whose position outside of the boundary is statistically significant ($p$-value $<0.05$), with error bars showing standard deviation and the size of the circles inversely proportional to the $p$-value. The means and variances used for the graphs shown in (B) and (F) can be found in S2 Data with the individual data points in S3 Data. The data underlying the graph in (E) can be found in S1 Data. RNAi, RNA interference.

in aggregates remain of unknown function (CG7785 (SPRYD7 in humans), CG16890 (FRA10AC1), CG14105 (TTC36), and CG18812 (GDAP2)), although mutation of GDAP2 in humans causes neurodegeneration, consistent with a role in quality control [60]. More is now known about 2 of the hits. CG4050 is a mammalian ortholog of TMTC3, one of a family of ER proteins recently shown to be O-mannosyltransferases; deletion of TMTC3 causes neurological defects [61,62]. CG5885 is the ortholog of the SSR3 subunit of the TRAP complex that also showed reduced wing size; in mammalian cells, the TRAP complex is up-regulated by ER stress [58]. These hits are consistent with reports that ER stress can increase cytosolic protein aggregation [63].

## Contribution of unknome genes to resilience to stress

Genomes have evolved to deal with many environmental stresses, and again, these are processes poorly investigated by traditional genetic approaches. We therefore tested resilience to stress, following knockdown of the unknome set. To quantify the viability of large numbers of flies, individual flies were arrayed in 96-well plates, and the plates maintained on a "flywheel" that rotated them under a camera every hour (Fig 4C and S1 Video). Viability was indicated by movement between images, allowing time of death to be determined with an accuracy of +/− 1 h (Fig 4D and 4E). We applied this method with 2 challenges likely to be associated with different cellular resilience mechanisms: amino acid starvation and oxidative stress.

**Resilience under starvation.** Under conditions of amino acid deprivation, knockdown of 8 of the unknome test set significantly prolonged survival (Fig 4F). Seven of these genes remain of unknown function, but interestingly, 5 have orthologs in other species whose localisation or interactions suggest that they have roles in the endosomal system. Thus DEF8, the mammalian ortholog of CG11534, has been reported to interact with Rab7 [64,65], and TMEM184A (CG5850) has been reported to act in the endocytosis of heparin [66]. In addition, the mammalian orthologs of CG4593 and CG9536 (CCDC25 and TMEM115) are Golgi-localised proteins of unknown function, and the yeast ortholog of CG13784 (ANY1) has been found to suppress loss of lipid flippases that act in endosome-to-Golgi recycling [67,68]. Our identification of this cluster of genes with related functions suggests that defects in endocytic recycling can prolong survival in starvation, possibly by altering autophagy or by reducing signalling from receptors that promote anabolism. The other 2 genes that improved starvation resilience when knocked down have no known function in any species, with loss of CG31259 (TMEM135) causing mitochondrial defects, and nothing reported for CG3223 (UBL7) [69,70]. One gene, CG15738 (NDUFAF6), caused an increased susceptibility to starvation, and it has been found to be an assembly factor for mitochondrial complex I, whose loss compromises viability [71].

**Resilience under oxidative stress.** Resistance to oxidative stress was tested with paraquat, an insecticide widely used to elevate superoxide levels in *Drosophila* [72,73]. There was considerable variability in the survival times, but 11 genes gave a statistically meaningful increase in resistance (Fig 4F). Most of these genes remain unknown, but 3 have since been reported to have functions related to oxidative stress signalling. The mammalian ortholog of CG4025 (DRAM1/2) is induced by p53 in response to DNA damage and promotes apoptosis and autophagy [74]. The mammalian orthologs of CG13604 (UBASH3A/B) are tyrosine phosphatases that repress SYK kinase, an enzyme reported to help protect cells against ROS, with superoxide activation of *Drosophila* Syk kinase signalling tissue injury [75–77]. Finally, the ortholog of CG3709 in archaea has tRNA pseudouridine synthase activity, but the human ortholog PUS10 has been reported to be cleaved during apoptosis and promote caspase-3 activity, thus its loss may slow apoptic cell death [78]. Of the other 8 hits, 5 remain poorly characterised, 1 is involved in mitochondrial function and so may reduce ROS production, and 2 are involved microtubule function with no clear link to superoxide responses. Although further validation will be required, these 5 genes seem good candidates to have a role in mitochondria or ROS-response pathways.

## Contribution of unknome genes to locomotion

Metazoans benefit from having a musculature under neuronal control. We therefore addressed the possibility of neuromuscular functions by testing the role of the unknome set of genes in locomotion, using the iFly tracking system in which the climbing trajectories of adult flies are quantified by imaging and automated analysis (Fig 5A) [79,80]. Climbing speed declines with age, so the assay was performed at both 8 days and 22 days post eclosion. Climbing speeds are inevitably somewhat variable, even in wild-type flies, but nonetheless 6 genes were statistically significant outliers when assayed after 8 days (Fig 5B). Two of these genes remain poorly understood, and for 3 of the others recent work indicates a role in muscle or neuronal function. These include CG9951, whose human homolog CDCC22 has been recently found to be a subunit of the retriever complex that acts in endosomal transport. Missense mutations in CDCC22 causing intellectual disability [81,82]. The human ortholog of CG13920 (TMEM35A) is required for assembly of acetylcholine receptors [83]. Finally, CG3479 is the gene mutated in the *Drosophila outspread (osp)* wing morphology allele, and is expressed in muscle, with one of its 2 mammalian orthologs (MPRIP) being been found to regulate actinomyosin filaments [84,85].

## Validation of fertility screen hits by gene disruption

Analysis of gene function by RNAi can be confounded by off-target effects. We therefore used CRISPR/Cas9 gene disruption to validate selected hits from 2 of the phenotypic screens. From the fertility screens, 3 male steriles and 1 female sterile were selected for genetic disruption. Of the male hits, CG10064 and CG6153 were both confirmed as being required for male fertility (Fig 6A to 6D). CG10064 is a WD40 repeat protein, and mutation of its human ortholog, CFAP52, results in abnormal left-right asymmetry patterning, a process known to depend on motile cilia [46,86]. CG6153 comprises a PITH domain that is also found in TXNL1, a thioredoxin-like protein that associates with the 19S regulatory domain of the proteasome through its PITH domain [87,88]. Males lacking CG6153 made morphologically normal sperm, but they did not accumulate in the seminal vesicle, the organ in which nascent sperm are stored prior to deployment, suggesting that they have limited viability (Fig 6E to 6J). Neither CG6153 nor its human ortholog PITHD1 are testis specific, and, indeed, orthologs are also present in non-ciliated plants and yeasts, suggesting that the protein has a role in an aspect of proteasome

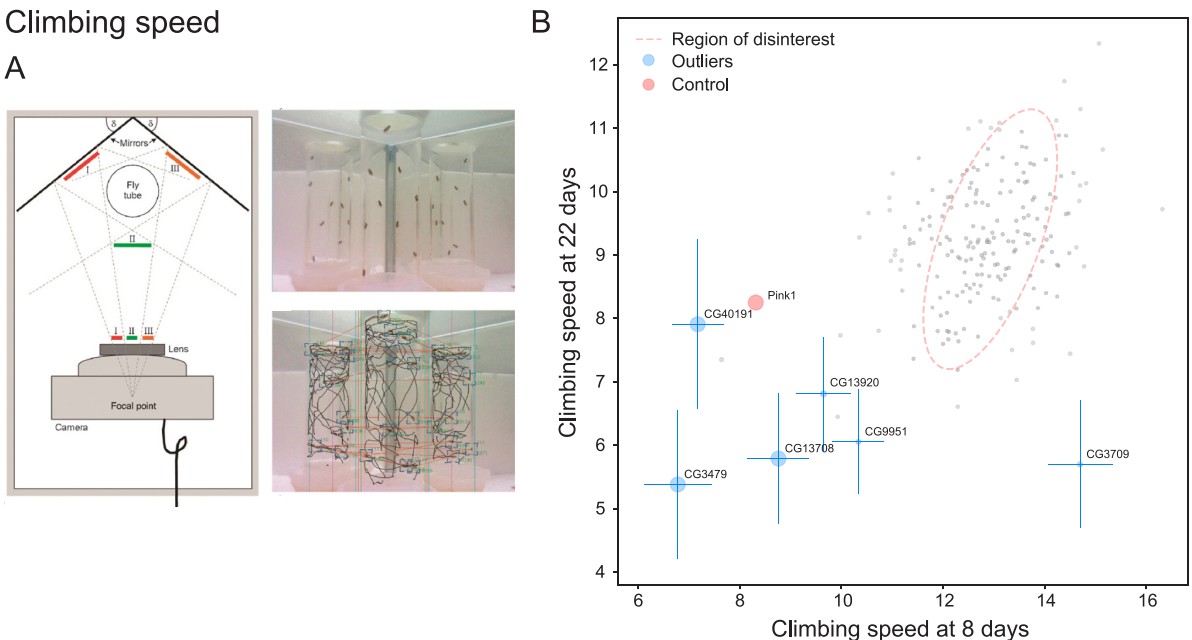

**Fig 5. Testing the unknome set of genes for roles in locomotion.** (A) iFly tracking system for automatic quantitation of *Drosophila* locomotion (reproduced from Kohlhoff and colleagues [80]). *Drosophila* are knocked to the bottom of a glass vial and placed in an imaging chamber that allows viewing from 3 angles and their climbing tracked automatically. (B) Plot of the mean climbing speeds of fly lines in which the unknome set of genes has been knocked down by RNAi, and the speeds for each line were determined after 8 days or 22 days post eclosion. Loss of the Parkinson's gene Pink1 affects climbing speed and it was included as a control [115]. Dotted lines indicate an outlier boundary set at 90% of the variation in the dataset, with the genes named being those whose position outside of the boundary is statistically significant with a *p*-value <0.1, with error bars showing standard deviation and the size of the circles inversely proportional to the *p*-value. The means and variances used for the plot shown in the figure can be found in S2 Data with the data points in S3 Data. RNAi, RNA interference.

biology that is of particular importance for maturing viable sperm. Recent work on mouse PITHD1 indicates it has a role in both olfaction and fertility [89,90]. The other male sterile hit, CG16890 (FRA10AC1), and the female sterile hit, CG8237 (FAM8A1), did not show reduced fertility when disrupted and presumably represent off-target RNAi effects (S3 Fig).

## Wing size hit CG11103 is a regulator of Notch signalling

Knockdown by RNAi of gene CG11103 (TM2D2 in humans) caused alterations in the growth of the wing (Fig 3D and 3E). When CG11103 was removed with CRISPR/Cas9, mutant females and males were viable without any obvious phenotypes, but females were completely sterile (Fig 7A and 7B). Eggs laid by mutant females were fertilised but failed to develop, and cuticle preparations and antibody labelling of the pan-neuronal marker Elav showed a hyperplasia of nervous system at the expense of the epidermis (Fig 7C–7G). This phenotype is characteristic of defects in the highly conserved Notch signalling pathway that is required in the *Drosophila* embryo to determine/specify the neuroblasts that give rise to the CNS in a process called lateral inhibition. CG11103 contains a TM2 domain that comprises 2 putative transmembrane domains connected by a short linker [91]. The function of this domain is unknown, but it occurs in 2 related proteins in *Drosophila*, and all 3 of the fly proteins have human orthologs (Fig 7B). Interestingly, one of these, *almondex*/CG12127, was identified as a gene required for

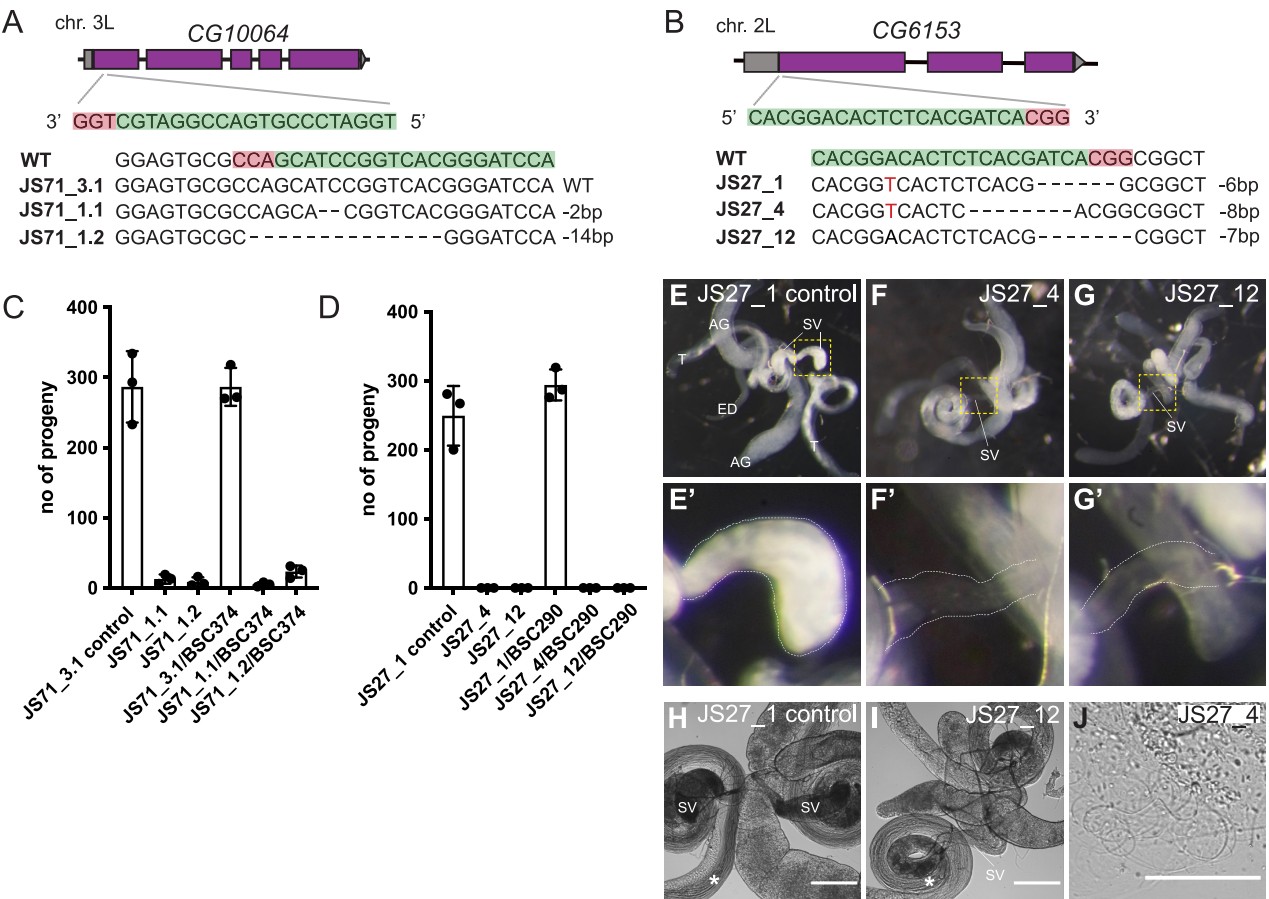

**Fig 6. Validation of RNAi male sterility phenotypes using CRISPR/Cas9 gene disruption.** (A, B) Schematics of the genomic locus of candidate genes, position of CRISPR target sites and mutant alleles analysed. (C, D) Assessment of male fertility of mutants (homozygous and over a deficiency). The graphs show mean values +/− SD of the number of progeny produced by mutant males. Three crosses with 5 wild-type virgins and 3 mutant males were analysed for each genotype. Wild-type males or males carrying in-frame mutations were used as controls. Where possible, alleles covering both alternative reading frames were analysed. (E–G) Widefield fluorescent micrographs of male reproductive systems of control and JS27/CG6153 mutants expressing Don Juan-GFP to label sperm. Mutants exhibit empty seminal vesicles, (E'-G') show zoomed regions of seminal vesicles from E–G (yellow dashed squares). (H–J) Widefield phase micrographs of reproductive systems of control and mutant males. Sperm are produced in both (asterisks), suggesting that sperm are made in the mutant but does not survive. Note that some mutant sperm gets into the ejaculatory duct (J). AG, accessory gland; ED, ejaculatory duct; SV, seminal vesicle; T, testis. Scale bars, 200 μm (H, I), 100 μm (J). The data underlying the graphs shown in the figure can be found in S1 Data. RNAi, RNA interference.

Notch signalling in embryos, although its role remains unclear [92]. The third related gene, *CG10795*, is also of unknown function, so we knocked it out with CRISPR/Cas-9 and discovered that it too showed phenotypes indicative of a severe defect in Notch signalling (Fig 7H–7L). Thus, all 3 proteins are required for a cellular process essential for embryonic Notch function, and recently, a similar conclusion was independently made by others [93]. All 3 human TM2D proteins were hits in a recent genome-wide screen for defects in endosomal function [94], and endosomes play a critical role in Notch signalling. Further work will be required to determine the precise role of these proteins, and how it relates to wing growth, but their likely role in endosomal function, combined with the existence of related TM2 domain proteins in bacteria and archaea, suggest fundamental roles in cell function rather than an exclusive role in Notch signalling.

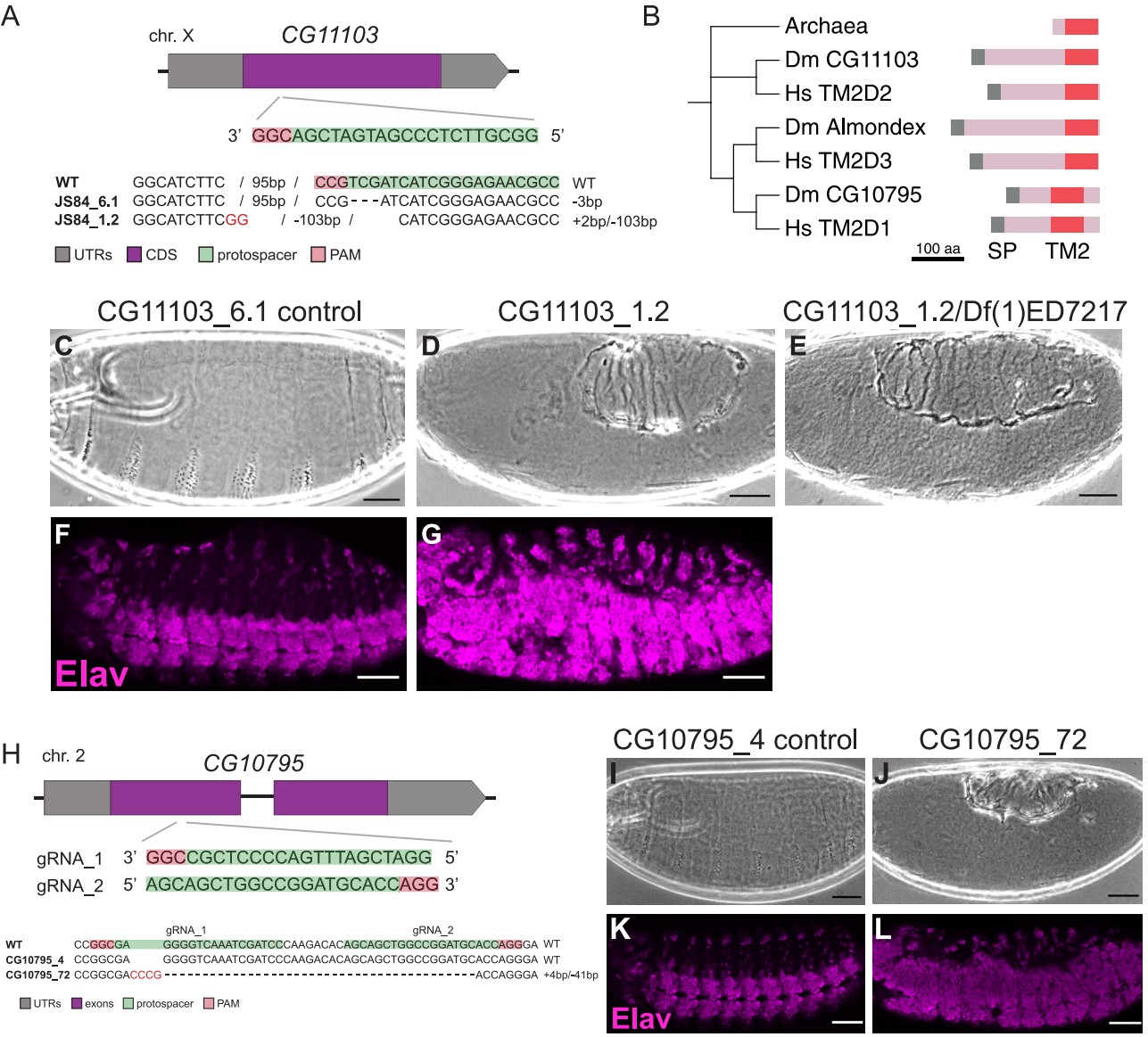

**Fig 7. Investigation of wing growth hit *CG11103* using CRISPR/Cas9 gene disruption.** (A) Schematic of the genomic locus of candidate *CG11103*, position of the CRISPR target site and the mutant allele analysed. Flies carrying an in-frame mutation were used as control. (B) Gene tree for TM2 domain proteins in humans and *Drosophila*, with an archaeal TM2 protein as an outlier. Tree built using sequence of TM2 domains alone using T-Coffee. A fourth TM2 domain protein is present in *Drosophila* and humans (Wurst/DNAJC22) which has additional TMDs and a DNAJ domain and appears to play a role in clathrin-mediated endocytosis [116]. (C–E) Cuticle phenotypes of embryos laid by control females and mutant females (homozygous or over a deficiency). (F, G) Micrographs of embryos laid by control females and homozygous mutant females stained against the pan-neuronal marker Elav. Scale bars: 50 μm. (H) Schematic of the genomic locus of *CG10795*, position of CRISPR target sites and the alleles analysed. Flies without an indel were used as control (*CG10795_4*). (I, J) Cuticle phenotypes of embryos laid by control or mutant females. (K, L) Micrographs of embryos laid by control or mutant females stained for the pan-neuronal marker Elav. Scale bars: 50 μm.

Taken together, this genetic validation data confirms that the RNAi screening approach, despite its known caveats, has given accurate phenotypic information for at least a substantial subset of the hits from our RNAi screens of the unknome set of genes.

## Discussion

The totality of scientific knowledge represents the summed activity of numerous individual research groups, each focusing on specific questions whose selection is influenced by many factors, some scientific and some more socially determined [7]. The latter set of factors includes issues like a preference for the relative safety, sociability, and kudos available when working in well-established fields, but is also strongly influenced by funding mechanisms. These usually aim to address societal needs but are subject to subjective assessment, historical precedent, and political pressures. In particular, the need to justify proposed research with reference to an established body of work, and preliminary data, may restrict investigation into truly unknown areas. Putting it more positively, there is potential for scientific progress to be accelerated by identifying situations where questions are being inadvertently and unjustifiably neglected. To quote James Clerk Maxwell "Thoroughly conscious ignorance is the prelude to every real advance in science." We have thus directly addressed here an area of long-standing concern: that biological research largely ignores less well known, but potentially important, genes [2,4,6,7]. Our results provide further evidence that this concern is well founded.

Our approach has been to develop an Unknome database. This has confirmed previous observations that poorly understood genes are relatively neglected; we also find that this problem is persisting even though there has been some progress in assigning functions to some of these genes. Recent developments in exome sequencing have allowed the identification of novel components of pathways whose genes give a well-defined set of disease symptoms, as has been seen with the cilia proteins identified from patients with ciliopathies [42,95]. In addition, the advent of the CRISPR/Cas9 system has enabled screens that cover whole genomes [17,96]. However, such screens are typically performed in cultured cells and hence cover only a subset of biological processes, and can also miss genes that have closely related, and thus functionally redundant, paralogs [97].

We used the Unknome database to select 260 genes that appeared both highly conserved and particularly poorly understood, and then applied functional assays in whole animals that would be impractical at genome-wide scale. Using 7 assays, designed to interrogate defects in a broad range of biological functions, we found phenotypes for 59 genes, in addition to the 62 genes that appear to be essential for viability (S2 Table and S4A and S4B Fig). Our approach relied on RNAi, but when 7 of the hits (corresponding to 6 genes) were retested with CRISPR/Cas9 gene disruption, we could validate 4. This is also a reminder that studies in model organisms such as *Drosophila* still have the scope to provide insight into unstudied human genes. The use of RNAi to knockdown candidate genes is powerful in this context because it allows for tissue-specific knockdown moreover, the likely incomplete loss of function achieved by RNAi can allow essential genes to reveal otherwise hidden hypomorphic phenotypes. Conversely, we note that as CRISPR approaches become ever more streamlined and sophisticated, future exploitation of the Unknome database can realistically use CRISPR technology to investigate functions of unknown genes.

An important primary conclusion of our work is that these uncharacterised genes have not deserved their neglect, a conclusion strengthened by a variety of other studies published during the protracted course of our studies, again revealing important functions for unknown genes. Again, this highlights the gradual shrinking, albeit slowly, of the unknome. Perhaps, most significantly, our database provides a powerful, versatile, and efficient platform to identify and select important genes of unknown function for analysis, thereby accelerating the closure of the gap in biological knowledge that the unknome represents. In practical terms, the Unknome database provides a resource for researchers who wish to exploit the opportunities associated

with unstudied areas of biology. Such endeavours will of course carry some risk as the outcome will be uncertain, and indeed, there is evidence that junior scientists are less likely to become principal investigators if they work on genes that have received little previous attention [7]. One approach may be collaborative efforts between labs to share resources and risk, and indeed, such an approach has recently been suggested by a consortium of proteomics groups [98].

Thinking about how to evaluate ignorance of gene function guided our bioinformatic approach to selecting of a set of genes small enough for complex phenotypic screening in a whole animal. At a broader level, we believe that acknowledging and evaluating ignorance is an important factor in decisions about the relative priority given to addressing the remaining fundamental questions in biology, versus translating and exploiting what we already know. However, ignorance can only have value if it can be meaningfully measured. Developing the Unknome database highlighted a couple of issues that affect our assessment of the state of knowledge of gene function. First, our approach relied on identifying orthologs from major organisms used for biological research. Although current methods for ortholog identification work well, there is still scope for improvement [24,25].

Secondly, our approach relied on the comprehensive and systematic annotation of gene function by the Gene Ontology (GO) Consortium [21,22]. Thus, another issue that arises from our work is that the current rapid rate of genome sequencing has required that most annotation is now automated rather than manual. This has led to the development of powerful methods to add functional annotation based on similarities to genes from other species [99]. However, such methods aim to cumulatively add annotation rather than remove disproven conclusions or address contradictions, which requires time-consuming manual curation. Moreover, increasing numbers of functional annotations are based on phenotypes from high-throughput screens for genetic phenotypes or protein–protein interactions, both of which are prone to generating false positives [100]. Thus, genes inevitably accrete annotations over time, some of which may be wrong, contradictory, or superficial but have little prospect of being corrected in the foreseeable future. As a result, the admirable aim of adding new gene annotation carries the risk of inadvertently obscuring our understanding of what is genuinely unknown.

An illustration of this problem is the gene CG9536 (TMEM115 in humans). This protein has been annotated as having endopeptidase activity based on distant sequence similarity to the rhomboid family of intramembrane proteases. However, CG9536, and its relatives in other species, lack the conserved residues that form the active site in rhomboids, and thus the only thing that can be currently concluded about the function of CG9536 is that it is almost certainly not a protease [101]. A more extreme case is htt, the *Drosophila* ortholog of huntingtin. This was not in the unknome test set because the extensive study of the role of huntingtin in human disease has led to many preliminary suggestions of function that have resulted in annotations linked to transcription, transport, autophagy, mitochondrial function, etc., and yet, the current consensus is that huntingtin's precise cellular role remains uncertain [102].

In conclusion, we find that accurately evaluating ignorance about gene function provides a valuable resource for guiding biological studies and may even be important for determining strategies to efficiently fund science. We have developed an approach to tackle directly the huge but under-discussed issue of the large number of well-conserved genes that have no reliably known function, despite the likelihood that they participate in major and even possibly completely new areas of biological function. We hope that our work will inspire others to define and characterise further the unknome and also to seek to ensure that gene annotation has the support and technology to preserve and recognise true ignorance.

## Materials and methods

### Construction of the Unknome database

The protein sequence data that we considered corresponds to the reference UniProt Proteomes [https://www.uniprot.org/proteomes/] used by the latest PANTHER database and includes human and 11 model organism species: *A. thaliana*, *C. elegans*, *D. rerio. D. discoideum*, *D. melanogaster*, *E. coli* (K12), *G. gallus*, *M. musculus*, *R. norvegicus*, *S. cerevisiae*, and *S. pombe* [26,103].

The Unknome database aggregates relevant information from the listed sources and provides a default knownness score for each protein and protein family (cluster) and can be recompiled in a few hours. Here, PANTHER provides the protein family information, via a group of UniProt IDs, that can be combined with selected information from UniProt entries, including protein sequence, GO terms, PubMed citations, species, gene name(s), and cross-references to species-specific databases.

The GO terms present in each UniProt entry were automatically provided by the Gene Ontology Annotation (GOA) database [https://www.ebi.ac.uk/GOA], based on GO release 2022-09-19 [22]. Evidence terms from the OBO Foundry are employed by GO [104], and in the Unknome database, they were weighted according to their evidence codes using the following default values: EXP; 0.8, IDA; 0.8, IPI; 0.8, IMP; 0.8, IGI; 0.8, IEP; 0.8, ISS; 0.5, ISO; 0.5, ISA; 0.5, ISM; 0.5, IGC; 0.3, RCA; 0.6, TAS; 0.9, NAS; 0.6, IC; 1.0, ND; 0.0, IEA; 0.0, NR; 0.0, IRD; 0.0, IKR; 0.0, IBA; 0.5, IBD; 0.5 (see http://geneontology.org/docs/guide-go-evidence-codes/ for a full description). After weighting, they were summed to generate a knownness score for each protein. The knownness score for the family defined by PANTHER is the maximum score among all the protein members present in the human and model organism list.

All protein GO terms linked in the database were dated according to when they were first linked with the UniProt entry, so as to be able to track the historical change of knownness. Though this information is not directly accessible within UniProt entries, the GOA database makes this information available via GAF format files at ftp://ftp.ebi.ac.uk/pub/databases/GO/goa/UNIPROT/. Note that this information only covers current entries and so annotations made in the past that were subsequently removed are not included in analyses of the change in knownness.

The Unknome is presented with a web interface at the URL http://unknome.org, with the entire database available to download as SQLite Version 3 files. This website is constructed using the Python module Django and provides views on the underlying database with easy filtering by knownness. In particular, the site displays the change over time in knownness for each protein cluster and lists the GO terms associated with each member of the cluster, along with their dates. The web site also makes all data available for download, from individual protein sequences to the whole SQL database file.

### *Drosophila* genetics

Hairpin RNAi stocks for the Unknome set were from the KK library of the Vienna Drosophila Resource Centre (S1 Table). During the course of our studies, it was reported that the stocks in this library have the transgene in one of 2 sites in the genome (the annotated locus 40D or the non-annotated site 30B), and insertions at 40D can cause lethality when the guide RNA is expressed [32,33]. PCR analysis with the previously used diagnostic primers was applied to 360 of the 365 lines, with the 5 remaining lines being lethal when expressed and so not included in any of the functional screens. This PCR analysis revealed that 98 of the 360 lines have the transgene in the problematic 40D site, a frequency of 27%, comparable to the 23% (9/39) and 25%

(38/150) found previously. All but one of these 98 lines gave a lethal or semi-lethal phenotype when crossed to the ubiquitous da-GAL4 driver (S1 Table).

Expression of the RNAi hairpins was driven with either the ubiquitous driver da-GAL4 driver, or with tissue-specific drivers: en-GAL4 (wing), bam-GAL4-VP16 (male fertility), MTD-GAL4 (female fertility), and GMR-GAL4 (proteostasis in the eye). UAS-Dicer-2 was included in all cases except for the 2 fertility screens as this has been found to improve the efficiency of RNAi [105]. For the proteostasis screen, the driver line also contained UAS-Httex1--Q46-eGFP [59]. In the lethality screen, those crosses that produced no adult progeny were defined as "lethal," while those where the progeny reached the pharate stage but the majority could not hatch, and those that did failed to expand wings and did not survive, were "semi-lethal."

For validation using CRISPR/Cas9, the following fly stocks were used: nos-phiC3; attP40 (DBSC #25709), nos-phiC3;;attP2 (DBSC #25710), CFD2 [106], TH_attP2 [107], Df(1)ED7217 (DBSC #8952), Df(2R)BSC268 (DBSC #26501), Df(2L)BSC812 (DBSC #27383), Df(2L) BSC290 (BDSC #23675), Df(3L)BSC374 (BDSC #24398). Spermatids and sperm were labelled with Don Juan (dj)-GFP [108].

## Fertility

Fertility was monitored using competitive assays, in which 1 red-eyed fly expressing the RNAi and 1 white-eyed w1118 fly were placed with 4 w1118 flies of the opposite sex. For male fertility, the Bam-Gal4 driver was used in combination with Dicer, and for female fertility, MTD-Gal4 was used without Dicer. RNAi stocks for the controls were from the VDRC: vret (GD 34897) and Ref1 (KK 10447). The flies were allowed to mate for 7 days, transferring to fresh vials every 2 to 3 days. After 7 days, the parental generation was removed and all progeny that emerged from the vial were counted, with eye colour used to determine the parent of each. Flies from the RNAi parent were separated, imaged, and quantified using Fiji image analysis platform [109], with a custom macro (https://github.com/tjs23/unknome). Individual data for both males and females were used to calculate means and the variances errors for the graphical plot (S2 and S3 Data).

## Wing growth assay

The genes in the unknome set were knocked down in the posterior half of the wing by using an engrailed-GAL4 driver combined with UAS-dcr-2. For each cross, at least 10 independent wings were collected and mounted on a slide under a coverslip in 50% glycerol/PBST. Images obtained with a 5× objective were analysed using a Fiji macro to contrast the veins from the rest of the wing (https://github.com/tjs23/unknome), and then, the areas of specific inter-vein regions in the anterior and posterior halves were determined. Individual data for each stock was used to calculate means and the variances errors for the graphical plot (S2 and S3 Data).

## Proteostasis assay in the eye

To interrogate the handling of misfolded proteins, a GFP fusion to part of huntingtin with a polyglutamine repeat was expressed in eyes, and the number of GFP-positive aggregates determined [59]. UAS-Httex1-Q46-eGFP was expressed in the eye along with the RNAi using GMR-Gal4. One eye from at least 10 males per genotype was imaged after 18 days at 25˚C, using 3 males per independent cross. GFP-positive aggregates were quantified with Fiji using a custom macro that determined the area of the eye and then scored aggregates that were either smaller or larger than 50 pixels (https://github.com/tjs23/unknome). Individual data for each

stock was used to calculate means and the variances errors for the graphical plot (S2 and S3 Data).

## Survival under stress

To measure lifespan under stress, we developed an automated system for following viability over many days. Flies were placed in 96-well plates and photographed every hour with image analysis then used to identify when the flies stopped moving. To prepare the plates, nitrogen-free fly food was placed at the bottom of each well (8 g agar, 50 g glucose, and 5 g pectin per litre with 0.25% nipagin, antibiotics, and 4 ml/litre propionic acid as preservative). To assay oxidative stress, the same food was used with the addition of 7.5 mM paraquat. Adult male flies were subdued with $CO_2$ and single flies placed in each well of the 96 well, with the plate sitting on ice to prevent escape before the plate was full. The plate was then sealed with gas permeant film. To image the plates over time, they were placed on a circular rotating platform and moved under a camera to be imaged every hour, with 3 such platforms or wheels arranged in a stack. At least 200 adults were assayed for each genotype, and custom Python scripts used to align the images of each plate and then track the movement of the flies in each well (https://github.com/tjs23/unknome). Lifespan was defined as the time point after the last change in position of the fly in the well. Individual data for both starvation and ROS conditions was used to calculate median survival times and the variances errors for the graphical plot (S2 and S3 Data).

## iFly climbing assay

The climbing speed of flies was measured using the iFly tracking system in which a single camera and mirrors are used to follow the movement of flies in a vial [75,76]. The RNAi stocks for the unknome set were crossed to the ubiquitous daughterless-Gal4 driver, and progeny collected at 8 days and 22 days post eclosion. The Pink1 control RNAi stock was from the VDRC (KK 109614). To follow locomotion, 8 flies were placed in a vial that was tapped to collect them at the bottom, and then placed in the iFly apparatus for filming over 30 s, with this repeated 3 times. Locomotion velocities were then determined using the iFly tracking software [80]. Individual data from both 8 days and 22 days was used to calculate means and the variances errors for the graphical plot (S2 and S3 Data).

## Summary of statistical methods

The general approach we took is as follows, with full details provided as Supporting information (S1 Text). We first modelled the distributions of the experimental results relating to each of the phenotypes under consideration parametrically. We thus formalised the goal of identifying outlying genes as identifying outlying sets of parameters corresponding to genes for each of the different phenotypes. Our approach involved 3 steps. First, we performed a regression to obtain estimates of the parameters for genes and an estimate of their variance–covariance matrix while controlling for batch and other effects. This was important because variability across batches was substantial for several of the phenotypes considered. The particular regression model used for this batch correction depended on the dataset.

The next step involved determining an outlier region. To do this, we transformed the parameter estimates so they more closely resembled a sample from a normal distribution such that an elliptical outlier region was appropriate. This transformation was often simply chosen as the identity, but in certain cases logistic transformations were used, for example. To describe how this region was determined, it will be helpful to fix the phenotype and write $\mu_1, \ldots, \mu_J$ for the unknown transformed parameters for the genes, where $J$ is the total number of genes

under consideration for that phenotype. Furthermore, let us write $\hat{\mu}_1, \ldots, \hat{\mu}_J$ for the corresponding (transformed) estimated parameters. Note that the $\mu_j$ were two-dimensional in most examples.

We modelled the $\mu_j$ as samples from a mixture of a normal distribution and a distribution of outliers and aimed to estimate the mean and variance matrix of this normal distribution to give the center and shape of the outlier region. The mean was estimated using a robust mean estimator applied to $\hat{\mu}_1, \ldots, \hat{\mu}_J$, such that the outlying genes did not influence the estimate. Analogously, we also obtained a robust estimate of the variance of the $(\hat{\mu}_j)_{j=1}^J$ to better reflect the variance of the bulk of the $(\mu_j)_{j=1}^J$. We then employed a bootstrap approach [110] to adjust this variance estimate to account for the sampling variability of the $(\hat{\mu}_j)_{j=1}^J$: The raw robust variance would be an overestimate of the corresponding quantity for the true transformed parameters.

Given the final mean and variance estimates, we took our outlier region to be the complement of the elliptical contour of a normal density with this mean and variance with a size such that the probability of falling outside the region was either 0.05 or 0.1, depending on the dataset. Note that in the cases where the parameters $\mu_j$ were one-dimensional, the ellipse was simply an interval. Finally, we performed a bootstrap hypothesis test for each gene $j$ with the null hypothesis being that $\mu_j$ falls within the outlier ellipse. We thus obtained $p$-values for each gene quantifying the evidence that it is an outlier according to the data. Note that this measure incorporates how outlying $\hat{\mu}_j$ is, but importantly it also takes into account the fact that $\hat{\mu}_j$ is a noisy estimate of the true $\mu_j$. These $p$-values were then corrected for multiple testing using the Benjamini–Hochberg procedure [111].

## CRISPR/Cas9-mediated knock-out

CRISPR target sites were chosen using the CRISPR Optimal Target Finder (http://targetfinder. flycrispr.neuro.brown.edu/). pCFD3 was used for BbsI-dependent gRNA cloning (http://www. crisprflydesign.org/) [106]. gRNA transgenics were generated for all candidate genes using BDSC stocks #25709 or #25710, depending on the chromosomal location of the target gene. To generate indels, transgenic gRNA lines were crossed to either CFD2 or TH_attP2. DNA microinjections were performed by the University of Cambridge Department of Genetics Fly Facility. For generation of *CG10795* mutants, gRNAs were cloned into pCFD3, and plasmids injected into CFD2 embryos. Stable stocks were generated to recover indels for all candidate genes. For genotyping, single males were collected and the genomic DNA was isolated using microLYSIS-Plus (Clent Life Science). Diagnostic PCRs followed by sequencing identified indels. Antibodies were not available to check protein levels, and so for those genes where we did not observe a phenotype, it is formally possible that residual or truncated protein was to blame.

## Fertility assays on CRISPR/Cas9 mutants

To check male fertility, crosses with 5 Oregon R wild-type virgins and 3 mutant males were set up for each genotype. Crosses were kept at 25°C and knocked over twice. The total number of offspring was counted for all crosses and the mean +/− SD was plotted for each genotype. Deficiencies uncovering the candidate genes were used to check for potential off-target effects. To check female fertility, 3 crosses with 5 mutant virgins and 3 Oregon R wild-type males were set up for each genotype and processed in the same was as for male fertility. A deficiency uncovering *CG8237* was used to check for potential off-target effects.

### Analysis of *CG11103* and *CG10795* embryonic phenotypes

Overnight egg collections (at 25˚C) from *CG11103* and *CG10795* mutant females and males were kept at 25˚C for 48 h. Dead embryos were dechorionated and mounted in Hoyer's medium. Slides were kept at 65˚C for at least 24 h and widefield images obtained with a Zeiss Axioplan microscope. For examination of Elav expression, overnight egg collections from *CG11103* and *CG10795* mutant females and males were dechorionated with bleach and fixed using 4% formaldehyde. Embryos were devitellinised using n-Heptane/Methanol. Embryos were washed in PBS/0.1% Tween20 and blocked in PBS/0.1% Tween20 plus 5% BSA. Mouse anti-Elav (1/20; DSHB) were added over night at 4˚C, and then, embryos washed in PBS/0.1% Tween20. Donkey anti-mouse Alexa 488 (Fisher Scientific) was added and left for 2 h at RT. Embryos washed in PBS/0.1% Tween20 and mounted in Vectashield containing DAPI (Vector Laboratories) and imaged on a Zeiss LSM 710 confocal.

### Analysis of male seminal vesicles in *CG6153* mutants

Testes from 3 to 5 days old adult males were dissected in PBS and then either directly transferred onto a slide with Schneider's medium to take live images using a Zeiss 710 confocal microscope or fixed in 4% paraformaldehyde for 30 min at RT. PFA was then removed and the testes washed in PBT 0.1% Tween 20. Images were taken on a Zeiss stereomicroscope and a Nikon digital camera.

## Supporting information

**S1 Fig. Features of the Unknome database.** (A) Illustration of the interface in the Unknome database that can be used to weight GO annotations depending on the type of evidence. The settings shown are the default weightings that were used to generate an unknome gene set. (B) Clusters in the unknome that contain at least one human protein ranked by knownness, showing the distribution of proteins that are defined by Pfam as being in an uncharacterised protein family (UPF) or containing a domain of unknown function (DUF). The data underlying this graph can be found in S1 Data.
(EPS)

**S2 Fig. Trends in knownness.** (A) Change in the distribution of knownness of the 13,421 clusters that contain at least 1 protein from humans or the 11 model organisms. (B) Number of Gene Reference into Function (NCBI GeneRIF) annotations added per year since 2010 to the human genes in each of the 7,515 clusters that contain at least 1 human gene, ranked into deciles based on knownness in 2010. The best-known clusters in 2010 have received the most annotation in subsequent years. (C) Mean number of GO terms added to human-containing clusters per year for clusters ranked in deciles of knownness. The number of Process and Function GO terms added to all the genes in a cluster was summed and a mean determined for each year for all clusters in that centile. (D) Conservation in model organisms of human proteins in clusters as ranked in intervals of current knownness. (E) Mean number of species in each human-containing cluster as ranked in intervals of current knownness. Species are those in PANTHER 17.0, and better-known clusters tend to be present in a larger number of species. The data underlying the graphs shown in the figure can be found in S1 Data.
(EPS)

**S3 Fig. Testing of RNAi sterility hits using CRISPR/Cas9 gene disruption.** (A) Schematics of the genomic locus of candidate JS353/CG16890, position of CRISPR target sites and mutant alleles analysed. (B) Assessment of male fertility of CRISPR mutants in JS353/CG16890

(homozygous and over a deficiency). The graphs show mean values +/− SD of the number of progeny produced by mutant males. Three crosses with 5 WT virgins and 3 mutant males were analysed for each genotype. WT males or males carrying in-frame mutations were used as controls. Alleles covering both alternative reading frames were analysed. (C) Schematic of the genomic locus of candidate JS40/CG8237, position of the CRISPR target site and the mutant allele analysed. (D) Assessment of female fertility of mutants (homozygous and over a deficiency). The graph shows mean values +/− SD of the number of progeny produced by mutant females. Three crosses with 5 mutant virgins and 3 WT males were analysed. WT males and males carrying an in-frame mutation were used as controls. The data underlying the graphs shown in the figure can be found in S1 Data.
(EPS)

**S4 Fig. Graphical summary of the phenotypic screens.** (A) All genes that were analysed in the 7 phenotypic RNAi screens with those showing a phenotype in a screen indicated in red (see also S2 Table). For each screen, a few genes were omitted due to technical issues such as insufficient numbers of a particular cross being obtainable, or genes were analysed before they were found to be lethal and hence omitted from subsequent screens, and these are shown as blanks. The degree of conservation between each *Drosophila* protein and its human ortholog is indicated by the area of the circle shown. (B) Degree of amino conservation between the *Drosophila* proteins in the unknome set and their human orthologs, with the set that gave phenotypes (S2 Table), compared to those that did not. When there was more than 1 human ortholog in the cluster, the most closely related one was used. Relatedness calculated using the BLOSUM62 matrix. The data underlying the plot and the graph shown in the figure can be found in S1 Data.
(EPS)

**S1 Text. Supplemental text describing the statistical methods in depth.**
(PDF)

**S1 Table. List of *Drosophila* genes used for the unknome screen and the corresponding RNAi stocks.**
(XLSX)

**S2 Table. Genes whose knockdown gave significant effects in the functional screens.**
(XLSX)

**S1 Data. The data underlying the graphs and plots shown in Figs 1F, 2A, 2B, 4E, 6C and 6D, S1B, S2A–S2E, S3B, S3D and S4A–S4B Figs.**
(XLSX)

**S2 Data. Mean and variances from screens: Statistical analysis of batches assayed for each genotype in the functional screens, as used for plots in Figs 3A, 3D, 4B, 4F and 5C.**
(XLSX)

**S3 Data. Data points from screens: Data for individual flies from the batches assayed for each genotype in the functional screens as used to generate S2 Data.**
(XLSX)

**S1 Video. Representative time-lapse movie of a lifespan assay of flies in a 96-well plate.**
Frames captured every hour and played at 30 frames/second.
(MP4)

## Acknowledgments

We thank Damian Crowther for loan of the iFly tracking system, Sara Imarisio for advice on proteostasis assays, Tobias Klöpper for help with gene selection for screens, the LMB workshops for help with the system for lifespan measurements, Anna Parish for fly stock maintenance, and Manu Hegde for comments on the manuscript.

## Author Contributions

**Conceptualization:** Matthew Freeman, Sean Munro.

**Formal analysis:** Tim J. Stevens, Rajen D. Shah, Sean Munro.

**Funding acquisition:** Matthew Freeman, Sean Munro.

**Investigation:** João J. Rocha, Satish Arcot Jayaram, Nadine Muschalik, Sahar Emran, Cristina Robles.

**Methodology:** João J. Rocha, Satish Arcot Jayaram, Rajen D. Shah.

**Project administration:** João J. Rocha, Matthew Freeman.

**Software:** João J. Rocha, Tim J. Stevens.

**Supervision:** Matthew Freeman, Sean Munro.

**Validation:** Nadine Muschalik.

**Visualization:** Tim J. Stevens.

**Writing – original draft:** Sean Munro.

**Writing – review & editing:** Matthew Freeman, Sean Munro.

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
