## [Editor Report · Decision Letter 0]

17 Jan 2023

Dear Sean, 

Thank you very much for submitting your manuscript entitled "Functional unknomics: exploiting the value of ignorance in biological research" for consideration as a Methods and Resources Article by PLOS Biology.

Your manuscript has now been evaluated by the PLOS Biology editorial staff, as well as by an academic editor with relevant expertise, and I am writing to let you know that we would like to send your submission out for external peer review.

IMPORTANT: After discussions with the academic editor, we would like to consider your manuscript as a Short Report rather than a Methods and Resources Article. This would reframe the advance offered by the study as an analysis of genes with unknown function via a Drosophila screen, rather than focusing on the development of the Unknome database. We would be grateful if you could change the article type to a Short Report upon resubmission (see below).

Before we can send your manuscript to reviewers, we need you to complete your submission by providing the metadata that is required for full assessment. To this end, please login to Editorial Manager where you will find the paper in the 'Submissions Needing Revisions' folder on your homepage. Please click 'Revise Submission' from the Action Links and complete all additional questions in the submission questionnaire.

Once your full submission is complete, your paper will undergo a series of checks in preparation for peer review. After your manuscript has passed the checks it will be sent out for review. To provide the metadata for your submission, please Login to Editorial Manager (https://www.editorialmanager.com/pbiology) within two working days, i.e. by Jan 19 2023 11:59PM.

Kind regards,

Richard

Richard Hodge, PhD

Associate Editor, PLOS Biology

rhodge@plos.org

PLOS

---

## [Decision Letter · Decision Letter 1]

17 Feb 2023

Dear Sean,

Thank you for your patience while your manuscript "Functional unknomics: exploiting the value of ignorance in biological research" was peer-reviewed at PLOS Biology. It has now been evaluated by the PLOS Biology editors, an Academic Editor with relevant expertise, and by three independent reviewers. 

In light of the reviews, which you will find at the end of this email, we would like to invite you to revise the work to thoroughly address the reviewers' reports.

As you will see, the reviewers think the study is interesting well done but raise a few overlapping concerns for both the Unknome database and phenotypic RNAi screen sections of the manuscript. They ask that additional reporting details are provided for the screens, such as the positive/negative controls used and how many replicates were conducted, as well as providing the raw data in the Supplementary Information. Reviewer #1 provides a few suggestions that we agree would help to make the results more accessible to a broad audience, such as providing a graphical summary of the genes identified and comparing the results to other Drosophila hairpin screen databases. After discussions with the academic editor, we still feel your manuscript would be a good fit as a Short Report and we ask that the focus of the manuscript is not altered to highlight the Unknome database more than it currently is, or reducing discussions about the importance of studying unknown genes. 

Upon resubmitting the revised manuscript, we ask that you please reduce the number of main figures to 4 to fit with our Short Report format (https://journals.plos.org/plosbiology/s/what-we-publish#loc-Research-based-content). This could be achieved by either combining the main figures or by moving some of the figures to the Supplementary. 

Given the extent of revision needed, we cannot make a decision about publication until we have seen the revised manuscript and your response to the reviewers' comments. Your revised manuscript is likely to be sent for further evaluation by all or a subset of the reviewers.

**IMPORTANT - SUBMITTING YOUR REVISION**

*Re-submission Checklist*

*Published Peer Review*

*PLOS Data Policy*

*Blot and Gel Data Policy*

Kind regards,

Richard

Richard Hodge, PhD

Associate Editor, PLOS Biology

rhodge@plos.org

REVIEWS:

Reviewer #1: The manuscript by Rocha et al. provides two thrusts to facilitate studies of little investigated genes. In the first thrust they rank genes according to the extent of prior annotation. In the second thrust they provide functional insight into several little studied genes. 

The main novelties of the first thrust lie in the inclusion of annotations across diverse organisms, and in creating a tool (interactive database) that would allow other researchers to weight the impact of distinct annotations toward knownness according to their preference. The main novelties of the second thrust lie in the scope of the considered understudied genes and the diversity of phenotypic assays. Aside from the two thrusts, the manuscript consolidates the insight that several unstudied genes have some physiological relevance, and coins the phrase "unknomics" for a yet unnamed approached to targeted gene scholarship.

On the technical side, would like to note that GO can undergo changes across different years that remove annotations (Gillis et Pavlidis, 2013). Therefore, I suggest some extra words on whether temporal trends that are shown in the manuscript are based on the GO release of the indicated years, or based on a retrospective inspection of a current release. This might also affect the conclusion on knownness necessarily decreasing over years.

Further, GO annotations can themselves be informed by knowledge about orthologs. Thus, knownness score may be a composite of annotations and organisms with orthologs. To better frame conclusions, particularly on genes with low knownnness only being present in some other model organisms (Fig 2D), I would have found helpful a supplemental panel that compared knownness scores against the number of orthologs in other species.

Reproducibility could be increased by adding the release number of GO or download date to the methods section.

The main text and abstract could be more transparent in conveying the absolute number of hits of distinct assays. Similarly, the observed verification rate is around 50% (which is estimated from a small number of genes, and could thus in reality be even higher or lower given the statistical uncertainty around a small number of sampling events; also the verification rate well differ for some of the distinct assays used here). Thus, providing an uncorrected estimated of the share of genes with phenotypes (as for viability in the current abstract) could lead to an inflated perception of the magnitude of the findings. As it does not seem necessary for this manuscript that many hits have phenotypes, and as others have already reported on the importance of unstudied genes, some more moderate presentation would in my view be preferable as some readers might not be experienced with Drosophila hairpin screens.

Based on my own contribution to a genome-wide tissue-specific in-vivo Drosophila hairpin screen and the design of secondary and tertiary screens, I would similarly personally shy away from findings that seem derived from a single screen and/or biological replicates (experimental week, batch of food etc.). I am not advocating for more experimental work here, as there already is a lot of work that went into the experiments presented here, and the initial leads presented here seem very reasonable and useful to encourage and direct a further exploration of these genes. As the potential readership that extends beyond Drosophila specialists, would encourage an extra sentence or two that help readers not familiar with Drosophila how to engage with the specific screen results presented here. 

In terms of presentation, I greatly enjoyed the smart idea of marking the human gene symbol next to the Drosophila gene symbol as this will increase findability through search engines. However, it seems that for some genes this scheme has not been applied yet.

Fig 1E seems to indicate a "rich-get-richer" effect. An alternative presentation of the underlying data, which would promise to emphasize the value of unknomics and thus the present manuscript would be to test and show (very expected based on own side observations) that genes with fewer annotations will gain additional annotations at a slower absolute rate than well annotated genes.

I would find it helpful to see a graphical summary of genes vs. screens. For instance, a clustermap could show whether little investigated genes are pleiotrophic, or have very specific phenotypes. Similarly it would be interesting to see which share of the screened genes have no detected phenotype. Such a meta-analysis could possibly also be extended to see if certain pre-existing annotations (no matter how few) are informative on the phenotypes. Acknowledging that this manuscript seems to be under consideration for a brief report, presenting some of the present findings as a meta-analysis might also allow to shorten some of the text and main figures devoted to the specific screens.

Related to the above point of contextualizing findings, a comparison to GenomeRNAi, which aggregates Drosophila hairpin screens, or a comparison to the murine IMPC phenotype database, could help to convey the relative advance brought forward by targeted screens on under-investigated genes over the mining of existing data. To guide further studies of unstudied genes either outcome would be a useful insight.

Lastly, the manuscript appears to focus on genes with a low knownness score in Drosophila. This invites the possibility that those genes that have a phenotype in specific assays might already be well known in other organisms (including humans). While again either outcome would be useful for guiding future studies into unstudied genes, I would be thrilled to learn whether the value of functional unknomics lies in adding information to unstudied genes of model organisms, or whether findings on unstudied genes of model organism could (still) provide functional insight into unstudied human genes.

Reviewer #2: In this interesting manuscript, Rocha et al. pinpoint the importance of the unstudied genome (or "unknome") suggesting that poorly studied proteins are not necessarily less important than widely studied ones. To facilitate studies about unknown proteins, the authors have assembled a database of proteins from several organisms and assigned a knowledge value based on the number of previous studies on that specific protein. Then they have a tested in drosophila the function of 260 genes that were previously poorly studied and have identified phenotypes in several assays, ranging from fertility to starvation resistance. The authors should consider the following points:

-Please provide the raw data for all the screens that have been performed as supplementary material. Table S2 currently reports only a qualitative description of only the RNAi that scored in the different assays. This is insufficient: please provide numerical values for all the assays that have been done and are shown in Fig. 3-5 and for all the lines that have been screened, including those that did not score in the specific assay. Please also indicate how many times and with how many flies the assay was done. Please clearly specify what RNAi stock has been used. The stocks have been identified as "JS" stocks but this does not correspond to any official naming of the VDRC stocks or any other collection.

- Please provide information for the negative and positive controls that were used in each RNAi screen (currently there are some indicated only for some assays). I guess "Pink" used as control for the climbing speed is "Pink1"?

- Based on the screens performed, is there any correlation between the degree of conservation between drosophila and human and the probability of uncovering a phenotype?

- There is large-scale interactome data for drosophila, mice, human, and other organisms. The authors may want to consider integrating this into the unknome database. For this paper, is there any correlation between protein-protein interactions and uncovering a phenotype in the screens? For example, do the proteins that scored in the assays interact with previously-studied proteins involved in the same processes? 

Reviewer #3: This manuscript by Rocha, et al. uses extensive bioinformatics and statistical analysis to assign scores for orthologous protein clusters found in humans and at least one other model organism to raise awareness of uncharacterized and understudied proteins. The authors created a publicly available database where users can assess a so-called 'knownness score' for proteins of interest based upon multiple factors, including GO terms or experimental knowledge. A group of proteins with unknown gene function (>350) were subjected to RNAi analysis in Drosophila to assess possible function related to seven categories - viability, fertility, wing growth, response to stress, locomotion, and protein aggregation. The database was a Herculean amount of work and provides a valuable resource for the scientific community, especially if the website will be updated and maintained. Improvements in the manuscript are required, especially toning down the language that overstates some of the conclusions. The value of this paper is in the database, not necessarily the biological experiments.

(1) What is the timeframe for this study and/or development of the website? The phrase 'during the course of our studies' is used and it is unclear how long these efforts took. 

(2) Social aspects is described as a factor that may promote scientific bias. The meaning of this is not clear and likely not a valid argument without supporting data. Please remove.

(3) Figure 1F is not referred to in the manuscript.

(4) Many of the graphs would benefit from descriptive titles. For example, Figs 2A and S2A look similar, yet compare human vs non-human model organisms separately. Also true for tables such as 1E and 2C.

(5) da-Gal4 is a weak ubiquitous driver and should be noted as such. Likely more RNAi lines would be lethal if a stronger driver was used.

(6) Please include the rationale in the text for the controls utilized for fertility analysis and locomotion. Also include controls for Figures 4B and 5F. They provide a nice comparison for the effects seen.

(7) The axes for Figure 3D should be in a distance measurements, not pixels. 

(8) A better resolution picture is needed for Figure 4A as the current one looks grainy. Representative pictures for the positive hits in panel 4B should also be included.

(9) Is there confirmation of reduced mRNA and/or protein levels for any of the CRISPR/Cas9 mutants?

(10) The following statement 'Male flies lacking CG10064 produced motile sperm, but following mating they did appear to not persist in the female's sperm storage organ, the seminal receptacle' is not supported by any data.

(11) Is there published literature to support the statement 'suggesting a true hit rate of ~50%, a reasonable outcome for an RNAi-based approach.' Is it true that 50% of RNAi screens are hitting off-target? This seem unlikely and should be modified.

(12) The word 'ignorance' is used multiple times throughout the manuscript and is misleading. While the definition of ignorance is a lack of knowledge, the word implies uninformed or lack of education. As scientists, all of us want knowledge. Just because something has not been studied previously does not mean the community is ignorant. Please change this language throughout the manuscript, especially in the title. 

(13) The abstract (and elsewhere) states the present work demonstrates the importance of poorly understood genes. This isn't new knowledge or surprising data. It is understood that many uncharacterized genes likely have important functions. Please change this type of language throughout the manuscript. The focus should be on the website.

---

## [Editor Report · Decision Letter 2]

6 Jun 2023

Dear Sean,

Thank you for submitting your revised manuscript "Functional unknomics: exploiting the value of ignorance in biological research" for publication as a Research Article at PLOS Biology. This revised version of your manuscript has been evaluated by the PLOS Biology editors and the Academic Editor.

Based on our Academic Editor's assessment of your revision, I am pleased to say that we are likely to accept this manuscript for publication, provided you satisfactorily address the following data and other policy-related requests that I have provided below (A-F):

(A) We would like to suggest the following modification to the title: 

“Functional unknomics: in vivo screening of conserved genes of unknown function reveals the importance of poorly understood genes”

(B) You may be aware of the PLOS Data Policy, which requires that all data be made available without restriction: http://journals.plos.org/plosbiology/s/data-availability. For more information, please also see this editorial: http://dx.doi.org/10.1371/journal.pbio.1001797

-Supplementary files (e.g., excel). Please ensure that all data files are uploaded as 'Supporting Information' and are invariably referred to (in the manuscript, figure legends, and the Description field when uploading your files) using the following format verbatim: S1 Data, S2 Data, etc. Multiple panels of a single or even several figures can be included as multiple sheets in one excel file that is saved using exactly the following convention: S1_Data.xlsx (using an underscore).

-Deposition in a publicly available repository. Please also provide the accession code or a reviewer link so that we may view your data before publication. 

Figure 1F, 2A-B, 3A, 3D, 4B, 4E-F, 5C, 6C-D, S1B, S2A-E, S3B, S3D, S4A-B

(C) I appreciate that much of the underlying data that I have requested above is already provided in the S1 Data file. I would be grateful if this file could be annotated so it is made clear which specific figure panels relate to each dataset in the file. 

(D) In addition, I note that a S2 data file is referred to in the manuscript, but there does not appear to be a S2 data file in the File Inventory. I would be grateful if this could be provided upon resubmission.

(E) Please also ensure that each of the relevant figure legends in your manuscript include information on where the underlying data can be found, and ensure your supplemental data file/s has a legend.

(F) Please ensure that your Data Statement in the submission system accurately describes where your data can be found and is in final format, as it will be published as written there. 

We expect to receive your revised manuscript within three weeks. 

*Published Peer Review History*

*Press*

Best wishes,

Richard

Richard Hodge, PhD

rhodge@plos.org

PLOS

---

## [Editor Report · Decision Letter 3]

27 Jun 2023

Dear Sean,

On behalf of my colleagues and the Academic Editor, Ian Dunham, I am pleased to say that we can accept your manuscript for publication, provided you address any remaining formatting and reporting issues. These will be detailed in an email you should receive within 2-3 business days from our colleagues in the journal operations team; no action is required from you until then. Please note that we will not be able to formally accept your manuscript and schedule it for publication until you have completed any requested changes.

PRESS

Best wishes, 

Richard

Richard Hodge, PhD

rhodge@plos.org

PLOS
